# Using Probabilistic Approach to Evaluate the Total Population Density on Coarse Grids

**DOI:** 10.3390/e22060658

**Published:** 2020-06-14

**Authors:** Manal Alqhtani, Khaled M. Saad

**Affiliations:** 1School of Mathematics, College of Engineering and Physical Sciences, The University of Birmingham, Birmingham B15 2TT, UK; 2Department of Mathematics, College of Sciences and Arts, Najran University, Najran 11001, Saudi Arabia; kmalhamam@nu.edu.sa; 3Department of Mathematics, Faculty of Applied Science, Taiz University, Taiz 6803, Yemen

**Keywords:** sparse data, coarse grid, sampling, ecological monitoring

## Abstract

Evaluation of the population density in many ecological and biological problems requires a satisfactory degree of accuracy. Insufficient information about the population density, obtained from sampling procedures negatively, impacts on the accuracy of the estimate. When dealing with sparse ecological data, the asymptotic error estimate fails to achieve a reliable degree of accuracy. It is essential to investigate which factors affect the degree of accuracy of numerical integration methods. When the number of traps is less than the recommended threshold, the degree of accuracy will be negatively affected. Therefore, available numerical integration methods cannot guarantee a satisfactory degree of accuracy, and in this sense the error will be probabilistic rather than deterministic. In other words, the probabilistic approach is used instead of the deterministic approach in this instance; by considering the error as a random variable, the chance of obtaining an accurate estimation can be quantified. In the probabilistic approach, we determine a threshold number of grid nodes required to guarantee a desirable level of accuracy with the probability equal to one.

## 1. Introduction

One of the most important aims of an integrated pest management programme (IPM) is to provide an accurate estimation of the pest population size at the typical agricultural area. IPM can be defined as the incorporation of several different methods of pest management, which work together in a more sustainable way to protect food production from pest attacks [1]. This importance of obtaining an accurate estimation of pest population density has been discussed in [2]. After estimating the pest population abundance in an agricultural field, a decision of implementing a control action will be taken depending on the comparison of the estimation to some threshold value(s). For instance, if this estimate falls below the value of threshold, the decision in this case would be that no action is needed. However, according to the work in [3], if the estimation exceeds this threshold, then the decision is to implement a control action immediately. Many studies have proven that the application of pesticides is a widely used in order to reduce damage from pest attacks [4,5]. Thus, it can be seen from the above that precision is of utmost importance in order to arrive at the correct decision with regards to whether to implement a control action. If the true value of population density is known, then the decision can be made easily; however, this cannot be known, therefore reliable information with which to make an evaluation of population density is required.

There are several techniques that have been used to improve the measurement of accuracy. One method is to increase the number of samples in an agricultural field to calculate the value of pest abundance with a sufficiently large number of traps in statistical equation, given as
(1)$$M=\frac{1}{N}\sum_{i=1}^{N}f_{i},$$
where $f_{i}$ denotes the individual sample counts and *N* is the number of sample counts in an agricultural field [6], and then to consider it as the exact value of the pest population. In ecological research, the number of traps can be increased to hundreds per given agricultural area. However, in real-life pest monitoring programmes, the number of traps does not exceed twenty [6], and in some cases, it ranges between one and a few per typical agricultural area [7].

Numerical integration methods present an alternative to statistical methods of evaluation. In recent years, many studies have been carried out on these methods in order to present an overall view of their application to ecological problems [2,8,9,10,11]. These studies have shown that numerical integration methods are more reliable than standard statistical methods (Equation 1), which can be considered as a simple form of numerical integration. It has been demonstrated that advanced numerical integration methods tend to provide a more effective estimation of pest density. Generally, from a mathematical viewpoint, the problem of estimating the pest population size is considered as a problem of calculation of the following integral,
$$I=\int_{D}\int f\left(x,y\right)dxdy,$$
where the integrand f(x,y) is the pest population density.

Clearly, in ecological problems the function f(x,y) is only known at discrete points $f_{ij}=f\left(x_{i},y_{j}\right)$, and in this situation, it is impossible to find the analytical solution. In order to evaluate this integral, numerical integration methods must be used to approximate the solution [12,13,14,15,16].

In the 1d cases, integral (Equation 1) becomes
(2)$$I=\int_{a}^{b}f\left(x\right)dx,$$
where $f\left(x\right)\equiv f_{i},$
i=1,2,..,N. According to the work in [7], the general form of numerical integration can be obtained by the weighted sum as follows,
(3)$$I\approx I_{a}=\sum_{i=1}^{N}w_{i}f_{i},$$
where $w_{i}$, i=1,2,….,N rely on numerical integration methods.

The evaluation error has to be introduced as a result of replacing the exact population size *I* by approximation value $I_{a}$ where the relative error is defined as
(4)$$E_{rel}\left(N\right)=e\left(N\right)=\frac{\left|I-I_{a}\left(N\right)\right|}{\left|I\right|}.$$

It can be seen from the above that Formula (Equation 4) gives an accurate definition of the integration error only if the value of the integral *I* is known. However, in real-life applications, the exact value of the integral *I* is not available, and so Formula (Equation 4) can not be applied. Therefore, according to the work in [7], in order to overcome this obstacle, it is beneficial to present the asymptotic error estimate of the numerical integration method as follows,
(5)$$E\leq Ch^{q}sup_{x\in [a,b]}\left|u^{r+1}\left(x\right)\right|,$$
where C,q, and *r* are constants.

The evaluation accuracy will improve depending on the amount of available data. In other words, in order to achieve a more accurate estimation of the pest population size, the amount of data must increase accordingly [17]. In this case, the approximated solution will converge to the analytical solution. This means that accurate evaluation of the approximated solution requires a vast array of data to be available. However, in pest monitoring, a large amount of data requires more equipment and labour to perform the physical measurements. Furthermore, in terms of trying to generate simulated data, this will require a high computational effort. Therefore, the amount of data may be limited in various types of application and we have to deal with sparse data.

It has been shown in [9,17] that the standard evaluation technique does not work when the available data is sparse because of insufficient information in data collected on coarse grids. Another approach must be applied when dealing with high aggregation density distribution under various conditions. One common example of biological invasion which represents high aggregation density distribution, is that spreading the pest species at the beginning of invasion from a small area to reach as time progresses the entire domain. Within an agriculture field, the pest population is aggregated to a single subregion, and outside that subregion the density is zero. From a mathematical viewpoint, this pattern of distribution is known as a peak function. It is beneficial to make decision about the application of pesticides before the density become dominant over the whole agriculture field. In order to protect the production of the agriculture field, timely and accurate evaluation of the total number of pest insects at the beginning of invasion is very important. In this problem, the exact location of the peak is not known previously, as a result the application of numerical integration methods is hampered. Therefore, in order to overcome this obstacle, the number of traps must be increased uniformly over the entire domain instead of installing traps locally. However, in reality this procedure has failed to work with peak function due to insufficient information about the density function. It was discussed in [10,11] that the whole peak may be completely missed when peak is located between nodes of coarse grid. Consequently, the accuracy of evaluation the pest density can be affected severely and negatively. Therefore, instead of applying standard evaluation methods, a probabilistic approach is used to evaluate the total population size on coarse sampling grids. Recent cross-disciplinary research has accelerated in the field of computational and numerical algorithms, including methods for solving real-world ordinary differential equations, numerical integrals, and optimisation, which can be expressed as estimation algorithms. They are utilised to approximate the value of a variable, numerical, or stochastic integral, the solution of a differential equation, or the location of a maximum, as well as to optimise a multi-objective algorithm. Such techniques utilise principles of statistical inference and generalised probability theory. As such, they have been applied to these concepts in computer science, most notably artificial intelligence and modern machine learning.

Understanding and developing numerical techniques such as learning algorithms can be powerful in a wide range of multi-disciplinary research areas and industries. Combining such platforms encourages cross-fertilisation of research in computation, probability, and numerical methods. The interface of research between probabilistic and numerical methods has been applied to diverse and wide-ranging areas such as the Internet of Things, the development of algorithms for neural networks used in self-driving vehicles, and intelligent analysis systems, such as big data, stochastic methods in finance, and Monte Carlo simulation [18,19,20,21].

The paper is presented as follows. In Section 2, we will consider the evaluation error as a random variable and calculate probabilistic characteristics of the total population size. In Section 3, we will use the standard approach to calculate the mean and standard deviation of random variables. The probability of accurate evaluation at random location of the first grid node on a sampling grid will discuss in Section 4. We consider 2d probability analysis at regular and random locations of first grid node with ecological example in Section 5 and Section 6. In Section 7, we conclude our experience with probabilistic approach in the problem of accurate evaluation of the total population size.

## 2. Characteristics of the Probabilistic Approach for Higher Aggregated Population Density

In this section, the probability approach will be used to illustrate that when the number of grid nodes is very small, the accuracy is probabilistic rather than deterministic (i.e., achieving an accurate estimation becomes a matter of chance) [2]. Let us consider a peak function which represents an early stage of patchy invasion [2]. The population density function over interval D=[a,b] will be presented as a single peak function located within sub-interval $D_{u}=\left[x_{I},x_{II}\right]$, and the density will be zero elsewhere. Therefore, a highly aggregated density distribution u(x) can be defined as
(6)$$u\left(x\right)=\left\{\begin{array}{cc} f(x)>0 & \mathrm{for}x\in [x_{I},x_{II}]\\ f(x)=0 & \mathrm{otherwise} \end{array}\right..$$

The maximum of the peak is only obtained from the construction $x^{*}=\frac{\left(x_{I}+x_{II}\right)}{2}$ and it is assumed to be located within this construction. An example of a 1d peak function is shown in Figure 1a. The location of the peak must be considered randomly within a truncated sub-interval $\left[a_{I},b_{I}\right]$ taken from the original interval [a,b] (i.e., $a_{I}=a+\epsilon <x^{*}<b-\epsilon =b_{I}$). The restriction imposed on the location of the peak function is to prevent any important information on the population density being missed if the peak is located at the endpoints of the original interval [a,b] or at the boundaries of domain.

A standard example of a peak function is given by substituting f(x) in Equation (Equation 6) as follows,
(7)$$f\left(x\right)=\frac{1}{\sigma \sqrt{2\pi }}exp\left(-\frac{{\left(x-x^{*}\right)}^{2}}{2\sigma^{2}}\right),$$
where the location of the peak maximum $x^{*}$ is a random variable within the truncated sub-interval and σ is the standard deviation. The definition of the peak width as a function of the standard deviation is represented as follows [22],
(8)δ=6σ.

Now, let us generate peak function by using Formula (Equation 7) over interval [0,L] where L=300, σ=8, and the locations of peak will be changed randomly within $\left[a_{I},b_{I}\right]=\left[40,260\right]$ where ϵ=40 At a very small number of grid nodes, and when the function of highly aggregated density distributions is considered, the accuracy of the estimate cannot be determined and the integration error is dependent on the location of a random point $x^{*}$. Then, the error becomes a random variable as a result.

The generated function will be integrated over the interval [0,300] and a series of an increasingly refined number of regular grid nodes N=3,4,5,…., by applying statistical rule (Equation 1). Then, the relative error (Equation 4) will be computed to assess the accuracy of the method used, by using the criterion
(9)E≤τ,
where τ is a fixed tolerance number. The condition (Equation 9) must hold, where in ecological applications a good level of accuracy lies between 0.2≤τ≤0.5, in some cases, τ≈1 may be considered acceptable level of accuracy, where τ is the accuracy tolerance [23,24].

At each number of grid node, the computations will be repeated for $n_{r}=10^{4}$ realisations of the random variable $x^{*}$. Then, the probability p(h) of accurate numerical integration is computed as follows,
(10)$$p\left(h\right)=\frac{\hat{n_{r}}}{n_{r}},$$
where p(h) is the probability obtained numerically, *h* is the grid step size on the number *N* of grid nodes N=3,4,5,6,…., and $\hat{n_{r}}$ is the number of realisations for which the integration error satisfies the condition e≤τ where τ=0.25. We repeat the computations by increasing the number of grid nodes until the grid step size becomes $h\leq \frac{\delta }{2}$ then, the computations will be stopped. The peak function (Equation 7) is shown in Figure 1 for the peak width δ=48. The probability p(h) when the condition (Equation 9) of the function (Equation 7) holds is presented in Figure 1b.

The computation started from a grid of $N_{1}=3$ nodes and ended on a grid of $N_{15}=17$, nodes where the condition $h\leq \frac{\delta }{2}$ holds. It can be seen from Figure 1b that the integration error when integrating the function (Equation 7) on a very coarse grid of nodes will depend on the location of the peak, and the probability of achieving an accurate estimate is p(h)<1. Meanwhile, on a grid of N=13 nodes $h=\frac{1}{12}$, the error is deterministic and the error is e≤τ=0.25 no matter where the peak is located. Therefore, it follows from the results of integrating the function of highly aggregated density distributions that the integration error becomes a random variable, and the accuracy of numerical integration at a very small number of grid nodes is probabilistic rather than deterministic, even if an acceptable degree of accuracy is acquired.

The above discussion leads us to introduce a new type of grid to be added to the grid classification, which is defined as an ultra-coarse grid. Therefore, there are now three types of computational grid: fine grid, coarse grid, and ultra-coarse grid. With fine grids, the asymptotic error estimate (Equation 5) always holds. For coarse grid, the asymptotic error estimates do not hold; however, the error will be deterministic and it will satisfy the condition (Equation 9) for a chosen τ. In the ultra-coarse grid, the asymptotic error estimate does not hold and the accuracy is not deterministic. The accuracy can only described in terms of the probability of achieving an error within a prescribed tolerance τ. The probability satisfies p(h)<1 when the condition e≤τ holds for a chosen tolerance τ. Therefore, the threshold number $N^{*}$ of grid nodes must be determined, where at any number of grid node less than $N^{*}$, the probability is p(h)<1 and the computational grid is defined as an ultra-coarse grid. At the number of grid nodes $N\geq N^{*}$, the transition from ultra-coarse to coarse grid takes place and the error becomes deterministic.

Now, let us vary the value of standard deviation σ in the peak function (Equation 7) to investigate the impact of choosing different values of σ on the probability of achieving an accurate estimation. The peak function (Equation 7) will be considered and the value of σ will vary as follows σ=4,6,8,10,12,15. Then, at each value of σ, the above computations will be repeated. All values of threshold numbers of grid nodes, alongside their value of σ, are presented in Table 1.

It can be seen from Table 1 that the number of grid nodes required for the condition p(h)=1 to hold depends on the width of the peak function δ=6σ. By comparing the critical number of grid nodes required to achieve an accurate estimate at two different values of τ, it can be noted from Table 1 that there is an inverse relationship between peak width δ=6σ and $N^{*}$. For a wide peak, the critical number of grid nodes required is small, whereas this number becomes bigger at narrow peaks, as shown in Table 1 and Figure 2b. Furthermore, the number of grid nodes to fulfil the accuracy requirements in the probabilistic approach depends on the value of tolerance τ, where from Table 1 it can be seen that for a low level of accuracy τ=0.4, the required number of grid nodes is considerably less than the number required at τ=0.25 at the same values of peak width δ, as can be seen from Figure 3.

An instructive example is depicted in Figure 2a, where at fixed width σ=8 in (Equation 7), the probability function p(N) depends on the chosen tolerance τ: where at the high level of accuracy τ=0.1, the critical number of grid nodes $N^{*}$ required for p(N)=1 to hold is considerably bigger than the numbers required for τ=0.25 and τ=0.4, respectively.

We now consider the superposition of normal distributions given by the following equations,
(11)$$u_{j}\left(x,t\right)=\frac{w(t)}{P}\sum_{m=1}^{P}f_{j}^{m}\left(x\right),j=A,B,$$
where
(12)$$f_{j}^{m}\left(x\right)=\frac{1}{\sigma \sqrt{2\pi }}exp\left(-\frac{{\left(x-{\left(x^{*}\right)}^{m}\right)}^{2}}{2\sigma^{2}}\right).$$

This generates functions of four and eight peaks by substituting the value of *P* as P=4, or P=8. The previous computations, which are applied to the peak function (Equation 7) at several values of standard deviation σ=4,6,8,10,12,15, will be repeated for functions with four and eight peaks, respectively. The main aim of this variation in the number of peaks is to determine the effect of increasing the number of peaks on the probability of achieving an accurate estimate on a coarse grid nodes. For each value of σ at different numbers of peaks, P=1,4,8, in the function, the number of grid nodes required for the probability condition p(N)=1 to hold is recorded in Table 2. It can be noted from Table 2, by comparing the values of the number of grid nodes $N^{*}$ with different numbers of peaks, that when the number of peaks becomes P=8, the number of grid nodes $N^{*}$ required to achieve an accurate estimate is slightly less than that number $N^{*}$ required at number of peaks P=1,4, respectively, as shown in Figure 4. Therefore, when the number of peaks is increased we have a higher probability of obtaining an accurate estimate.

Let us now expand the population density function given by (Equation 6) in the vicinity of the location of the maximum $x^{*}$ as follows,
(13)$$u\left(x\right)=u\left(x^{*}\right)+\frac{1}{2}\frac{d^{2}u\left(x^{*}\right)}{dx^{2}}{\left(x-x^{*}\right)}^{2}+R\left(x\right).$$

In the vicinity of the peak, the remainder term R(x) can be ignored, so that the density peak function is given by the quadratic function as follows,
(14)$$u\left(x\right)\approx Q\left(x\right)=B-A{\left(x-x^{*}\right)}^{2},x\in \left[x_{I},x_{II}\right],$$
where $A=-\frac{1}{2}\left(\frac{d^{2}u\left(x^{*}\right)}{dx^{2}}\right)>0$, $B=u(x^{*})>0$. The maximum value of the peak coincides with the maximum value of the quadratic function, which is symmetric around the location of $x^{*}$. The quadratic function Q(x) must be always non-negative over sub-interval $\left[x_{I},x_{II}\right]$, i.e., $Q\left(x_{I}\right)=Q\left(x_{II}\right)=B-Ah^{2}\geq 0$. Then, the following condition must hold,
(15)$$h^{2}\leq \frac{B}{A}.$$

Let us now use the quadratic function given in Equation (Equation 14) to generate data over unit interval [0,1]. A uniform grid of *N* nodes is generated in the domain [0,1], as $x_{i+1}=x_{i}+h$ where the grid step size h=1/N−1, i=1,2,…,N−1. The population density in the vicinity of the peak will be considered, where for the interval $\left[x_{i-1},x_{i}\right]$, δ=2h is the width of the peak function. From Equation (Equation 15), the width of the peak function becomes $\delta =2\sqrt{\frac{B}{A}}$ and the roots are $x_{I}=x^{*}+\delta /2,x_{II}=x^{*}-\delta /2$. The grid step size *h* can be defined in terms of peak width as h=αδ, where the parameter α>1. The exact value of abundance *I* can be computed by treating the peak as a quadratic function, as follows:(16)$$I=\int_{x_{I}=x_{i-1}}^{x_{II}=x_{i+1}}Q\left(x\right)dx=2Bh-\frac{2Ah^{3}}{3}.$$

In order to ensure that the relative error satisfies the condition $E_{rel}\leq \tau$, in the quadratic function (Equation 14), the chosen value for the peak is always positive, i.e., A>0 and the grid step size *h* should satisfy the following condition $h\leq \sqrt{\frac{B}{A}}$.

Now, let us consider the quadratic function given by (Equation 14). As demonstrated in the previous section, the probability of achieving a sufficiently accurate estimate when modelling a peak function with normal distribution will be computed. The width of the peak is fixed as δ=0.03,0.06,0.12, and the location of peak $x^{*}$ is a random variable distributed uniformly over the interval [δ,1−δ], where the entire peak must be stationed within the unit interval [0,1]. By fixing the tolerance τ=0.25, the probability of achieving an accurate estimate at different values of peak widths can be calculated from Equation (Equation 10). The density of quadratic function (Equation 14) at δ=0.06 is shown in Figure 5a.

Figure 5b shows the probability p(h) of obtaining an accurate answer at peak width δ=0.06; from this figure, it can be seen that for a very small number of grid nodes, the probability is p(h)<1 and the integration error itself is a random variable with a high magnitude. The probability becomes equal to one at the critical number of grid nodes $N^{*}=23$. This required critical number of grid nodes $N^{*}$ increase as the value of the peak width decreases (i.e., a narrower peak) and it decreases as the peak width increases, as can be seen from Table 3. Table 3 presents the number of grid nodes required to achieve an accurate estimate of the quadratic function at different values of peak widths δ. The probability curves for approximating the quadratic function with peak widths δ=0.03,0.12 are displayed in Figure 6a,b. It can be deduced from Figure 5b and Figure 6a,b that at different values of peak width, the required value of probability when the number of grid nodes *N* is very small has not been achieved, and the integration errors become random variables with high magnitudes, due to inadequate information about the integrand quadratic function. Furthermore, the relative error when the number of grid nodes is very small tends to be large (i.e., $E_{rel}\to 1$), the accuracy in this case is probabilistic rather than deterministic, and achieving an acceptable level of accuracy will become a matter of chance in this case. By considering a peak as a quadratic function and the integration error as a random variable, the threshold number of grid nodes $N^{*}$ required to achieve a prescribed level of accuracy has been determined, where at each regular grid with $N\geq N^{*}$, an adequate level of accuracy holds.

## 3. Evaluating the Arithmetic Mean and Probabilistic Mean Population Density on Coarse Grids

Let us now consider the standard approach to calculating the mean and standard deviation of random variables $I_{a}$ given by (Equation 1), and the relative error *e* given by (Equation 4) on a grid of *N* nodes. Formulas (Equation 17) and (Equation 18) give us the arithmetic mean and standard deviation for uniform variables $I_{a},e$ as follows [6],
(17)$$\hat{\mu_{i}}=\frac{1}{n_{r}}\sum_{i=1}^{n_{r}}y_{i},$$
(18)$$\hat{\sigma_{i}}=\sqrt{\frac{1}{n_{r}}\sum_{i=1}^{n_{r}}{\left(y_{i}-\hat{\mu_{i}}\right)}^{2},}$$
(19)$$\mu_{i}=\sum_{i=1}^{n_{r}}p\left(y_{i}\right)y_{i},$$
(20)$$\sigma_{i}=\sqrt{\sum_{i=1}^{n_{r}}p\left(y_{i}\right){\left(y_{i}-\hat{\mu_{i}}\right)}^{2},}$$
where $n_{r}$ is the total number of realisations of the random variables, $y_{i}$ is a random evaluation of the total population size $I_{a}$, or a random evaluation error *e*, $p(y_{i})$ for a random $y_{i}$ is the probability density function, and the subscript *i* can be $I_{a}$ or *e*. According to the work in [6], the following formulas are true,
(21)$$|I-\mu_{I_{a}}\left(N\right)|⟶0\;as\;N⟶\infty ,\mu_{e}⟶0\;as\;N⟶\infty ,$$
(22)$$\sigma_{I_{a}}⟶0\;as\;N⟶\infty ,\sigma_{e}⟶0\;as\;N⟶\infty .$$
where $\mu_{I_{a}}$ is the arithmetic mean of a random variable of the total population size $I_{a}$, $\mu_{e}$ is the arithmetic mean of a random variable of the evaluation error *e*, $\sigma_{I_{a}}$ is the standard deviation of a random variable of the total population size $I_{a}$, and $\sigma_{e}$ is the standard deviation of a random variable of the evaluation error *e*. Let us now consider the quadratic function given in Equation (Equation 14) to generate data over the unit interval [0,1]. The width of the peak will be fixed as δ=0.03,0.06,0.12, and the location of peak $x^{*}$ is a random variable distribute uniformly over the interval [δ,1−δ], where the entire peak must be located within the unit interval [0,1]. The above computations are then repeated.

The density (Equation 14) at δ=0.06 is shown in Figure 5. The computation starts with a fixed number of grid nodes N=3, and then the number of grid nodes increases as $x_{i+1}=x_{i}+h$, i=1,2,…. At each number of grid nodes, the computations will be repeated until the following condition holds,
(23)$$\sigma_{e}\leq \tau =0.25.$$

Once the data has been obtained, the arithmetic mean of approximate integrals $I_{a}$ and numerical integration errors *e* will be computed by using Equations (Equation 17) and (Equation 18). It can be noted from Figure 7, Figure 8 and Figure 9, that at each value of peak width δ, the arithmetic mean of the approximated integral values tend to be very close to the exact value of the integral when the number of sample units is increased. Furthermore, at a narrow peak δ=0.03, the arithmetic mean converges slowly to the exact value of integral I=0.0045. It also has a slightly oscillating feature, as shown in Figure 8a. At a wider peak width, the approximate value $\mu_{I_{a}}$ converges to the exact values of integrals I=0.0360, and I=0.2880 faster when the width of the peak increases to δ=0.06, and δ=0.12, as depicted in Figure 7a and Figure 9a. It can be observed from Figure 7b, Figure 8b and Figure 9b that the sample mean $\mu_{e}$ tends to be zero when the number of grid nodes increases, and the approximation converges faster to reach zero when the width of peak in the quadratic function becomes wider, as shown in Figure 7b, Figure 8b and Figure 9b.

Table 4 introduces the number of grid nodes required for the condition (Equation 23) to hold at different peak widths for quadratic function (Equation 14) and when the probability p(N)=1. It appears from Table 4 that the number of grid nodes required for condition (Equation 23) to hold is slightly less than the number of grids required for condition p(N)=1 to hold.

For the population density given by (Equation 14), Figure 10 compares the arithmetic mean of error ${\hat{\mu }}_{e}$ and upper and lower bounds ${\hat{\mu }}_{e}\pm {\hat{\sigma }}_{e}$ with a tolerance of τ=0.25. When considering the peak function, the graphs of arithmetic mean error and bounds of error for different values of peak widths show the impact of peak width on the number of grid nodes needed to achieve the desirable accuracy ${\hat{\mu }}_{e}+{\hat{\sigma }}_{e}\leq \tau$. At the narrowest considered peak function δ=0.03, the number of grid nodes *N* required to fulfill the accuracy requirements is a significant N=41 points, as shown in Figure 10b, whereas this number decays gradually while increasing the width of the peak as δ=0.06 and δ=0.12 to be N=21 and N=11, as shown in Figure 10a,c. Therefore, in order to achieve a desirable accuracy, the grid has to be refined when dealing with a quadratic function that has different peak widths, and the degree of refinement depends on the width of the peak, as demonstrated above.

As we are dealing with continuous random variables, the probabilistic approach of computing the mean and standard deviation of variables $I_{a},$ or *e* given by (Equation 19) and (Equation 20) will be used. Let us consider the range of variable *y* to be $y\in [y_{i},y_{i+1}]$. The interval $\left[y_{i},y_{i+1}\right]$ will be divided into *M* sub-intervals, where the size of each sub-interval is $h_{y}=\left(y_{i+1}-y_{i}\right)/M$. The following approximation will be considered $y_{i+1}=y_{i}+h_{y}$, for i=1,2,…,M and $\bar{y}=\frac{y_{i}+y_{i+1}}{2}$, which is the midpoint of the sub-interval $\left[y_{i},y_{i+1}\right]$. Now, let us consider the value of variable *y* that denotes the approximate value of integral $I_{a}$. The range of $I_{a}$ is be $I_{a}\in \left[I_{a}\left(i\right),I_{a}\left(i+1\right)\right]$ and will be extracted to determine the frequencies $k_{i},i=1,\ldots ,M$ of having the value $I_{a}$. Then, the probability of having $I_{a}$ at each sub-interval will be computed using the following formula,
(24)$$p_{i}=\frac{k_{i}}{n_{r}},i=1,\ldots ,M,$$
where $n_{r}=10^{4}$ provides the number of random realisations. The probability of having $I_{a}$ in each sub-interval with a coarse number of grid nodes N=3 is very low, whereas the probability increases when dealing with fine number of grid points N=23, as shown in Figure 11a,b. In the coarse sampling grid, the range $\left[I_{a}\left(min\right),I_{a}\left(max\right)\right]$ of random variable $I_{a}$ is very small and only some realisations can be considered as a good approximation to the true population abundance I=1. The probability distribution in Figure 11a, is not quasi-uniform, therefore as a result of sampling procedure inaccurate realisations of random variable $I_{a}$ are more likely to appear. When increasing the number of grid points, the range of $I_{a}$ becomes bigger. The probability distribution computed on a grid of N=23 points has the range of $I_{a}$ within approximately 20% of the true population *I* as depicted in Figure 11b. This range is greater than the range of $I_{a}$ computed for the population density distribution (Equation 14) on the grid of N=3 points only.

Table 5 shows combinations of statistical approaches used to investigate the difference between the arithmetic mean and standard deviation of $I_{a}$, and the probabilistic mean and standard deviation. From Table 5, it can be noted that the probability p(N) at coarse grid nodes remains very small and the condition p(N)=1 is achieved when intensively refining the grid nodes. The threshold number of grid points required is $N^{*}=23$, and at any number of grid nodes less than $N^{*}$, the condition p(N)=1 does not hold. The arithmetic mean ${\hat{\mu }}_{I_{a}}$ converges to the exact size of population density *I*, as predicted in (Equation 21), but the probabilistic mean $\mu_{I_{a}}$ converges slightly quicker than ${\hat{\mu }}_{I_{a}}$ to the exact size *I*, as it accounts for random variables $I_{a}$, as shown in Table 5. It is known from (Equation 22) that the probabilistic standard deviation of $I_{a}$ tends to be zero while N⟶∞, as illustrated in Table 5. Furthermore, the difference between the exact value of the integral and the probabilistic mean of the integral tends to be zero when the number of grid nodes increases incrementally. Therefore, from these results, the following conclusion is true; the probabilistic approach is more effective than the deterministic approach when dealing with continuous random variables.

## 4. The Probability of Accurate Evaluation at Random Choice of the First Grid Node $x_{1}$ on a Sampling Grid

In previous sections, the probability of obtaining an accurate estimate was computed for regular computation grids and a random location of the maximum peak. The probability will now be computed for a quadratic peak function at a fixed location of the maximum peak, and the location $x_{1}$ on the uniform computational grid will be moved randomly within sub-domain [a,h], where *h* is the grid step size on a grid of *N* points and *a* is the left-hand endpoint of interval [a,b].

Let us consider the quadratic peak function given by (Equation 14) as the first test case for computing the probability of achieving a sufficiently accurate estimate, as demonstrated in the previous section. The width of the peak is fixed as δ=0.06, and the location of peak $x^{*}$ is fixed as $x^{*}=0.672$ within the unit interval [0,1]. Now the location of grid node $x_{1}$ is randomly moved over interval [0,h], where $h=\frac{1}{N-1}$ is the grid step size. The grid nodes must be stationed within the unit interval [0,1]; therefore, the last grid node $x_{N}$ will be excluded. We provide $n_{r}=10^{4}$ realisations of the random variable $x_{1}$ on a grid with a fixed location of the maximum peak at $x^{*}=0.672$. Then, the value of integral will be estimated by applying statistical method (Equation 1) at each realisation of $x_{1}$. The integration error will be computed by using the relative error (Equation 4). By fixing the tolerance τ=0.25, the probability of achieving an accurate estimate at different values of grid node $x_{1}$ is calculated by Equation (Equation 10), where in the probability equation $\hat{n_{r}}$ represents the number of realisations where the integration error is $E_{rel}\leq \tau$. The quadratic peak function (Equation 14) with $x^{*}=0.672$ is depicted in Figure 12a. Figure 12b shows the curve p(h) for the probability of obtaining an accurate answer at peak width δ=0.06. From this figure, it can be seen that at a very small number of grid nodes, the probability is p(h)<1 and the integration error is a random variable with the high magnitude. The probability becomes equal to one at a critical number of grid nodes $N^{*}=22$. This required critical number of grid nodes $N^{*}$ increases as the value of the peak width decreases (i.e., a narrow peak) and it decreases as peak width increases.

Table 6 shows the number of grid nodes required to achieve an accurate estimate of the quadratic peak function at different values of peak widths δ. The numbers of grid nodes required for the condition p(N)=1 to hold for the quadratic peak function (Equation 14) with peak widths δ=0.03,
0.06, 0.12, 0.18, 0.24 is displayed in Figure 12c. It can be deduced from Figure 12c that at different values of peak width, the required condition p(N)=1 does not hold when the number of grid nodes *N* is very small, and the integration errors become random variables with high magnitudes, due to inadequate information about the integrand quadratic peak function. Furthermore, the relative error when the number of grid nodes is very small tends to be quite large (i.e., $E_{rel}\to 1$), so the accuracy in this case is probabilistic rather than deterministic, and achieving an acceptable level of accuracy becomes a matter of chance in this case. The findings obtained at regular computational grid nodes with random locations of the maximum peak $x^{*}$ are almost identical to the results acquired at a fixed position of the maximum peak and a random location of grid node $x_{1}$, as shown in Figure 5b and Figure 12b.

Let us now reproduce some of the results of the 1d test cases introduced in [9,25] in order to compare them with the 2d test cases. We first consider the numerical test case that represents the 1d counterpart of the continuous front spatial density distribution [25], which is given by
(25)$$u\left(x\right)=\frac{1}{1+sin^{2}x},x\in \left[0,\pi /2\right].$$

The function u(x) shown in Figure 13a represents a spatiotemporal distribution spread throughout the domain from the left boundary of the region, where the population density is originally zero. The location of grid $x_{1}$ will vary with a fixed number of grid nodes *N* from $x_{1}=0$ to $x_{1}=h$, where *h* is the grid step size. The exact value of the population size in the domain is given by $I=\sqrt{2}\left(\pi /4\right)$. Grid nodes must be generated within the region [0,π/2]; therefore, the last grid node $x_{N}$ will be excluded. On a grid of N=3 points, when the value of $x_{1}$ increases from 0 to *h*, the approximated total population size $I_{a}$ is given by a continuous monotone function, as shown in Figure 13c. The range of variable $I_{a}\left(x_{1}\right)$ for $x_{1}\in \left[0,h\right]$ occurs within the interval $\left[I_{min},I_{max}\right]$, where $I_{min}=I_{a}\left(x_{1}=h\right)$ and $I_{max}=I_{a}\left(x_{1}=0\right)$. Therefore, any random location of $x_{1}$ will generate a randomly approximated $I_{a}$ taken from $\left[I_{min},I_{max}\right]$ [25]. It can be seen from Figure 13b that at the fixed tolerance τ=0.25 and a random location of $x_{1}$, the probability p(N)=1 of obtaining an accurate evaluation holds for a small number of grid nodes $N\geq N^{*}=5$. Increasing or decreasing the value of tolerance τ will decrease or increase this threshold number of grid nodes $N^{*}$, as demonstrated in [25].

In the sequence of regular number *N* grid nodes with $n_{r}=10^{4}$ random locations of $x_{1}$, the mean of variable error $\mu_{e}\left(N\right)$ and the standard deviation of variable error $\sigma_{e}\left(N\right)$ will be computed from Equations (Equation 19) and (Equation 20). The upper $\mu_{e}\left(N\right)+\sigma_{e}\left(N\right)$ and lower $\mu_{e}\left(N\right)-\sigma_{e}\left(N\right)$ bounds of the error will be computed in order to compare them with the tolerance τ. In order to approach the required level of accuracy, the following condition must hold,
(26)$$\mu_{e}\left(N\right)+\sigma_{e}\left(N\right)\leq \tau .$$

For Function (Equation 25), the condition (Equation 26) first holds at the number of grid nodes N=5, as shown in Figure 13d. The results obtained from Figure 13d prove that a good accuracy can be acquired with a coarse grid of N=5, where the upper bound of error falls below the tolerance τ=0.25. Therefore, for the distribution given by (Equation 25), at any grid node N≥5, the population density is considered to be a uniform distribution and the random variable *e* at any realisation will be within a desirable level of accuracy. Furthermore, $I_{a}$ provides a good evaluation of the total population density for any realisation on a grid of N≥5 points.

Consider now a function with a more sophisticated pattern that represents the 1d counterpart to the ecological phenomenon of highly aggregated spatial distribution [26,27].

Let us consider a single peak function introduced in the [25], which is given by
(27)$$u\left(x\right)=Aexp\left(-{\left(x-x^{*}\right)}^{2}/2\delta^{2}\right),x\in \left[0,L\right],$$
where A,δ,L, and $x^{*}$ are selected parameters. Figure 14a depicts the shape of Function (Equation 27) where A=1000, L=300, and δ=0.06. By increasing the value of $x_{1}$ continuously from 0 to *h* on a grid of N=3 points, the approximated total population size $I_{a}\left(x_{1}\right)$ is not a continuous monotone function of $x_{1}$, as shown in Figure 14c, and so it differs from the monotone function Figure 13c for function (Equation 25). It can be seen from Figure 14c that the range of random variable $I_{a}\left(x_{1}\right)$ is very large, and only for some realisations approximated values can provide a good approximation to the exact value of the population density I=1. The accuracy can be improved by increasing the number of grid nodes to, for instance N=33, to reduce the range of the random variable $I_{a}\left(x_{1}\right)$, as shown in work [25].

It can be seen from Figure 14b that at a fixed tolerance τ=0.25 and a random location of $x_{1}$, the probability p(N)=1 of obtaining an accurate evaluation holds for a number of grid nodes $N\geq N^{*}=24$, which is much larger than the number required for p(N)=1 to hold for function (Equation 25). Increasing or decreasing the value of tolerance τ will decrease or increase the probability of achieving an accurate evaluation for a fixed number of nodes *N*, respectively, as demonstrated in [25]. It has been illustrated in [25] that the required threshold number of grid nodes $N^{*}$ at τ=0.4 is $N^{*}=22$, and at τ=0.1 is $N^{*}=30$, which does not provide significant differences between them. The probability p(N) of the condition e≤τ holding depends on the shape of the population density distribution u(x), as proven in [9].

The upper $\mu_{e}\left(N\right)+\sigma_{e}\left(N\right)$ and lower $\mu_{e}\left(N\right)-\sigma_{e}\left(N\right)$ bounds of the error are computed in order to compare them with the tolerance τ. In order to approach the required level of accuracy, condition (Equation 26) must hold. For the function (Equation 27), condition (Equation 26) first holds at the number of grid nodes N=24, as shown in Figure 14d. The results obtained from Figure 14d prove that we require a high number of grid nodes to provide an acceptable level of accuracy when the upper bound of the error falls below the tolerance τ. Therefore, for the distribution given by (Equation 27), the probability density is considered as a nonuniform distribution of the random variable *e* on coarse grid nodes. On a grid of *N* nodes, a significant number of realisations of the random variable *e* may satisfy the condition e(N)≥τ, even when the upper bound of the error is very close to the tolerance curve τ. Furthermore, significant values of random variable $I_{a}$ will provide inaccurate approximations of the total population density on a grid of coarse points.

We want to extend the results in [25] and consider more computationally challenging examples. Let us now consider an exponent function with several peaks, by using the following equation,
(28)$$u\left(x\right)=\sum_{i=1}^{P}Aexp\left(-{\left(x-x_{i}^{*}\right)}^{2}/2\delta^{2}\right),x\in \left[0,L\right],$$
where *A*, *P*, δ, and $x^{*}$ are the chosen parameters. Let us fix the location of peaks $x_{i}^{*}$ as $x_{1}^{*}=89.3$, $x_{2}^{*}=187.6$, and $x_{3}^{*}=252.1$. At P=1 in (Equation 28), the single peak function (Equation 25) will be obtained and two-peak and three-peak functions are generated by fixing P=2,3, respectively, in (Equation 28). We now repeat the computations undertaken for (Equation 25) on the density distributions (Equation 28) of the two-peak and three-peak functions. The probabilities of obtaining an accurate estimation for in each case are shown in Figure 15b and Figure 16b. The shape of the two-peak and three-peak functions are depicted in Figure 15a and Figure 16a. It can be seen from Figure 15b and Figure 16b that the probability curves p(N) oscillate on a coarse grid node, and for the condition p(N)=1 to hold, a great deal of grid refinement is required. The probability of obtaining an accurate estimate of the total population size satisfies the condition p(N)=1 at a significant number of grid nodes, $N\geq N^{*}=27$, in both cases, as shown in Figure 15b and Figure 16b. The results obtained from (Equation 28) support the belief that the probability *p* depends on the shape of the population density distribution, as demonstrated by [9], and the probability function p(N) while dealing with exponent heterogeneous density distributions with more than one peak shows more sophisticated behaviour, as shown in Figure 15b and Figure 16b.

Let us consider heterogeneous population density with multiple peaks to investigate its convergence behaviour. The 1d function that represents a pattern of patchy density distribution is considered as follows [25],
(29)$$u\left(x\right)=4\pi^{2}xsin\left(A\pi x\right)cos\left(B\pi x\right)+C,x\in \left[0,1\right],$$
where parameters in (Equation 29) are taken as A=20.0,B=2.0, and C=50.0. The population density distribution given by Function (Equation 29) is shown in Figure 17a. Changing the value of $x_{1}$ continuously from $x_{1}=0$ to $x_{1}=h$ provides the range of variable $I_{a}\left(x_{1}\right)$, as shown in Figure 17c. It can be seen from Figure 17c that for an odd number of grid nodes N=3, 5, the shapes of variable $I_{a}\left(x_{1}\right)$ are approximate, which is different from the shape at even number of grid nodes N=4. Therefore, we expect that the probability of obtaining an accurate estimate of population density for random variable $I_{a}$ on coarse grids will show different behaviour for odd or even numbers of points. The shape of function p(N) is depicted in Figure 17b. From Figure 17b, it can be noticed that p(N) does not provide a monotone function, as presented in some previous test cases. Instead, it has oscillatory behaviour before hold the condition p(N)=1 at critical number of grid nodes $N\geq N^{*}=13$ then monotone behaviour will be acquired.

Now let us see how close the probability density distribution p(N) is to the evaluation error *e*, by computing the mean of error $\mu_{e}$ and the standard deviation $\sigma_{e}$ of random variable *e* on a sequence of regular grids. Figure 17d shows a comparison between $\mu_{e}$ and upper and lower bounds $\mu_{e}\pm \sigma_{e}$ for the tolerance τ=0.25. For the function with patchiness behaviour, the graph of mean error and bounds of error exhibit oscillating behaviour on coarse grids. Furthermore, as a result of increasing the number of grid nodes, the mean of error $\mu_{e}$ does not always guarantee an approximation close to the true population size, and the approximation of $\sigma_{e}$ does not necessary to decrease. However, an interesting observation is that the graph of p(N) shown in Figure 17b presents a strong form of synchronisation with the graphs of the mean of error $\mu_{e}$ and upper bound of error, as can be seen from Figure 17c, where the peaks of the probability p(N) obviously correspond to troughs of the upper bound $\mu_{e}+\sigma_{e}$.

The probability p(N) of obtaining an accurate evaluation will depend on the level of accuracy required, which is given by tolerance τ. It has been demonstrated in [9] that choosing different values of tolerance τ will present different behaviours of the probability function p(N) when a heterogeneous population density distribution (Equation 29) is considered. It has been shown in [9] that when the level of accuracy is low (i.e., for a large value of tolerance τ=0.4) the critical number of grid nodes required for p(N)=1 to hold is $N\geq N^{*}=2$. The critical number $N^{*}$ will increase as a result of increasing the level of accuracy to τ=0.2, where in this case $N^{*}=12$. The probability p(N) does not present a monotone function, but has oscillatory behaviour on a grid of coarse points, before the condition p(N)=1 holds at $N^{*}=12$. This oscillating behavior of p(N) becomes more obvious when the value of required tolerance decreases to τ=0.05, where the probability p(N) exhibits strongly oscillating behaviour on a coarse grid $N<N^{*}=14$. Therefore, it has been proven in [9] that at τ=0.4, the evaluation procedure becomes deterministic and the required accuracy of over 40% is obtained even on a grid of coarse points. Meanwhile, this evaluation procedure is probabilistic rather than deterministic on coarse grid nodes at τ=0.2,0.05 and a fine grid has to be used to fulfill the required level of accuracy. In the next section, we expand the results obtained in [9,25] by considering 2d density distribution for the purpose of comparison, as mentioned previously.

## 5. The 2d Probability Analysis

After analysing the probability of obtaining an accurate estimation for the 1d problems, our analysis is expanded to involve a more realistic 2d problem. Let us consider a peak function given by spatial distribution for the 2d counterpart of the normal distribution (Equation 7), where the function f(x,y) is given by
(30)$$f\left(x,y\right)=\frac{1}{\sqrt{2\pi \sigma^{2}}}exp\left(-\frac{{\left(x-x_{1}^{*}\right)}^{2}+{\left(y-y_{1}^{*}\right)}^{2}}{2\sigma^{2}}\right),\left(x,y\right)\in D_{u}.$$

The domain of interest is represented by the unit square D=[0,1]×[0,1]. In the equation (Equation 30), $\left(x^{*},y^{*}\right)$ provides information about the peak maximum and is chosen randomly. The sub-domain $D_{u}$ of the peak represents a circular disc of radius *R*, which is centered at $\left(x^{*},y^{*}\right)$. The peak width is given by δ=6σ=2R, which in the 1d distribution is fixed at δ=0.06. The 2d normal distribution given by (Equation 30) is depicted as a single peak in Figure 18.

We have determined the critical number of grid nodes required to satisfy p(N)=1 at $N^{*}=13$ for a 1d normal distribution (Equation 7) with δ=0.06. The computation will now be extended to obtain $N^{*}$ for a 2d normal distribution given by (Equation 30) with δ=0.06. Although we expect the probability graphs of p(N) to be the same for both 1d and 2d problems, the results obtained for the 2d normal distribution are shifted from the probability graph p(N) for the 1d normal distribution (Equation 7), as depicted in Figure 19a, where the critical number needed to gain p(N)=1 is $N_{2d}^{*}=24$, as can also be seen from Figure 19a. Table 7 shows how the critical number $N^{*}$ depends on the values of the chosen tolerances τ and σ in both 1d and 2d normal distributions. Furthermore, Table 7 supports the previous observation that the 1d probability graph for normal distribution shifts in the case of a 2d problem.

From Figure 19, it can readily be noticed that the width of the peak and the value of tolerance τ play important roles in deciding the critical number of grid nodes $N^{*}$ required to satisfy p(N)=1 in both dimensions for normal distributions. However, this critical number is higher in the case of 2d normal distributions, and it increases as the values of tolerances τ and σ decrease, as shown in Figure 19 and Table 7.

We now consider the superposition of 2d normal distributions to be given by:(31)$$u\left(x,y\right)=\frac{A}{4\pi \sigma^{2}}\sum_{j=1}^{P}exp\left(-\frac{{\left(x-x_{j}^{*}\right)}^{2}+{\left(y-y_{j}^{*}\right)}^{2}}{4\sigma^{2}}\right),\left(x,y\right)\in D_{u},$$
where $\left(x^{*},y^{*}\right)$ denotes the random location of peaks and *P* is the number of peaks. Let us first consider P=2, then the spatial population distribution density appears as two peaks, as shown in Figure 20a.

It can be seen in Figure 20b that the value of critical number $N^{*}$ for the condition p(N)=1 to hold for a 2d normal distribution of two patches P=2 is similar to the value for one patch P=1, where it is equal to $N^{*}$ = 24, for τ = 0.25, and $N^{*}=21$ for τ=0.4. Furthermore, the 1d normal distribution required fewer grid nodes to reach p(N)=1 than in the 2d case, as depicted in Figure 20b. The investigation was then expanded to involve 2d normal distributions with four and eight peaks P=4, and P=8, as shown in Figure 21.

It can be seen from Figure 22a that the probability graph p(N) of 1d and 2d normal distributions at fixed peak width δ and fixed tolerance τ=0.25 has been computed for different numbers of peaks. By inspecting Figure 22a, it can be readily seen that the critical number $N^{*}$ required to have p(N) depends clearly on the number of peaks, where this number at P=4 is bigger than $N^{*}$ at P=8. Moreover, the critical number needed for a 1d superposition of normal distribution is obviously lower than for the 2d problem. By varying the value of tolerance τ and number of peaks *P* for a 2d normal distribution with fixed δ=0.06, it can be seen from Figure 22b that a lower value of τ=0.25 with P=4 and requires more grid nodes $N^{*}=23$ to achieve p(N)=1, and this number decreases at P=8, to be $N^{*}=21$. When the degree of accuracy required decreases at τ=0.4, we expect that the critical number $N^{*}$ will decrease accordingly, as shown in Figure 22b, where it is equal to $N^{*}$ = 20, for P=4 and $N^{*}=17$ for P=8.

## 6. The 2d Probability Analysis at Random Locations of First Grid Node $\left(x_{1},y_{1}\right)$: Ecological Example

Let us now investigate the probability of obtaining an accurate evaluation of 2d ecological data generated from a system of diffusion—reaction equations given as
(32)$$\frac{\partial u(x,t)}{\partial t}=d\frac{\partial^{2}u}{\partial x^{2}}+u\left(1-u\right)-\frac{uv}{u+a}$$
(33)$$\frac{\partial v(x,t)}{\partial t}=d\frac{\partial^{2}v}{\partial x^{2}}+k\frac{uv}{u+a}-mv,$$
where the dimensionless parameters are introduced as $d=D/\alpha l^{2}$, k=κA/α, m=M/α, and a=H/L. Consider a spatial population density distributions in a squire domain [0,1]×[0,1], shown in Figure 23. The spatial distribution depicted in Figure 23 introduces a highly aggregated pattern of spatial distribution, which was obtained from Equations (Equation 32) and (Equation 33) for m=0.414 and t=450.

The exact value of the population density is obtained from (Equation 1) at the finest grid nodes N=1025×1025. Let us now consider a simulation procedure for sampling in domain D=[0,1]×[0,1]. The values of the population density are taken from the nodes of the regular grid sampling procedure, as taking sampling at the nodes of a regular grid is a common situation in ecological applications [26,28]. The computational grid nodes in the x-direction, $x_{i},i=1,\ldots ,N_{1}$ over interval [0,L] are generated as follows: $x_{i+1}=x_{i}+h_{1},i=1,\ldots ,N_{1}-1$, where we require that $x_{1}=a>0,x_{N_{1}}=b<L$ where *L* is the linear size of the domain and the grid step size $h_{1}=\left(b-a\right)/N_{1}$. Let us now consider a set of points in the y-direction, $y_{j},j=1,\ldots ,N_{2}$ over interval [0,L], generated as $y_{j+1}=y_{j}+h_{2},j=1,\ldots ,N_{2}-1$, where the grid step size $h_{2}=\left(d-c\right)/N_{2}$ for some 0<c<d<L. We require the location of the new grid node $x_{1}$ to be sufficiently close to the original grid node $x_{1}=0$ of the domain *D*; therefore, the conditions $0<x_{1}<h_{1},0<y_{1}<h_{2}$ must hold to make sure that the simulated sampling corner grid is located close to the corner in the domain *D*.

The location of grid node $\left(x_{1},y_{1}\right)$ is randomly moved over the domain $\left[0,h_{1}\right]\times \left[0,h_{2}\right]$, where $h_{1}=h_{2}=\frac{1}{N-1}$ and $N=N_{1}=N_{2}$ are the grid step size and the number of grid nodes in each direction, respectively. The grid nodes must be generated within the unit domain [0,1]×[0,1]; therefore, the last grid node $\left(x_{N},y_{N}\right)$ will be excluded. We provide $n_{r}=10^{4}$ realisations of the random variable $\left(x_{1},y_{1}\right)$. As we are dealing with simulated ecological data, and the density is only known at grid points, we therefore need to find the closest node to the new grid node $\left(x_{1},y_{1}\right)$ from the original computational grid nodes in the domain D=[0,1]×[0,1]. Once the data for the population density is extracted at the new first grid node $\left(x_{1},y_{1}\right)$, the value of the integral is estimated by applying the statistical method (Equation 1) at each realisation of $\left(x_{1},y_{1}\right)$. The integration error is computed by using the relative error (Equation 4). By fixing the tolerance τ=0.25, the probability of achieving an accurate estimate with $n_{r}=10^{4}$ random locations of $\left(x_{1},y_{1}\right)$ will be calculated from Equation (Equation 10).

It has been shown in [25] that, considering the bottom-left corner point of the sampling grid depicted in Figure 23 to be A=(a,c), a very accurate evaluation of the integration error $e\approx 10^{-2}$ is provided for the continuous front population density distribution. However, an inaccurate estimation of the integration error for the same sampling grid is provided for the patchy invasion. The integration error becomes much greater for a highly aggregated distribution. The degree of accuracy decreases dramatically from e≈1 to $e\approx 10^{-3}$ for the patchy invasion and the highly aggregated distribution respectively, when the location of grid *A* is moved slightly within the limited domain given above. Remarkably, choosing a random grid location $\left(x_{1},y_{1}\right)$ improved the degree of accuracy when dealing with all distributions of zero and non-zero patches. This procedure enables ecologists to increase the probability of catching high density value to gather these values with zero values, to eventually give us an accurate estimate of the population density.

The number of points required to achieve the condition p(N)=1 depends on the level of complexity within the ecological data. For the ecological distributions (Equation 23) the number of grid nodes required is $N^{*}=17$ in each direction to hold the probability condition, as depicted in Figure 24. A grid with a number of nodes below the value $N<N^{*}$ represent an ultra-coarse grid, where the evaluation of the integration error is probabilistic and is a matter of chance. Once the required number of grid nodes $N^{*}$ satisfies the probability condition, the integration error becomes deterministic and the asymptotic error (Equation 5) holds for a fine number of grid nodes.

In the sequence of regular number *N* of grid nodes with $n_{r}=10^{4}$ random locations of $\left(x_{1},y_{1}\right)$, the mean of variable error $\mu_{e}\left(N\right)$ and the standard deviation of variable error $\sigma_{e}\left(N\right)$ can be computed from Equations (Equation 19) and (Equation 20). The upper $\mu_{e}\left(N\right)+\sigma_{e}\left(N\right)$ and lower $\mu_{e}\left(N\right)-\sigma_{e}\left(N\right)$ bounds of error are computed to be compared with the tolerance τ. In order to achieve the required level of accuracy, the condition (Equation 26) must hold. It can be seen from Figure 25 that the condition (Equation 26) holds at N=17×17 the patchy invasion distributions, which is in fact the same number of grid nodes required to hold the probability condition p(N)=1. It can be concluded from this that, for ultra-coarse grid nodes, it is not possible to provide the information required to provide a reliable degree of accuracy; therefore, the integration error will be probabilistic rather than deterministic. Therefore, the probabilistic approach is used to estimate the size of population density for each ecological distribution.

## 7. Conclusions

A simple evaluation of the population density provides a deterministic level of accuracy when information about population density is sufficient. When dealing with a grid with a very small number of points, the information required to evaluate the population density is insufficient, which means that it is impossible to guarantee the required accuracy. Therefore, the probabilistic approach should be applied to evaluate population abundance. The probability *p* of achieving a sufficiently small error is used to assess accuracy, rather than considering the error itself; therefore, achieving the desired accuracy will be a matter of chance, rather than being deterministic, by considering the integration error as a random variable. The probabilistic approach requires series of statistical computations, e.g., the mean and the standard deviation, that in turn require repetitive and resource-heavy data collection techniques, which render the sampling procedure impossible.

For homogeneous density distributions, or density distributions slowly changing in space, any single realisation of a random variable obtained from the sampling procedure may possibly provide an accurate estimate of the true population density. In this case, any individual realisation can be considered to be a desired approximation of the population density, and the mean value will be close to the true population size. Moreover, at any realisation, the standard deviation value will tend to be small. However, when dealing with heterogeneous population density, the above observations are incorrect. It has been shown in this chapter that there is a significant difference between a single realisation and the true population size in heterogeneous distributions. The probability of producing an accurate evaluation of population density on ultra-coarse grids is low and is a matter of chance.

On ultra-coarse grids, the conventional approach of convergence analysis do not work, as it does not distinguish between the performance of different numerical integration methods. Therefore, the probability approach has been considered as a measurement of accuracy instead of conventional convergence analysis. A desirable level of accuracy is achieved if the condition p(N)=1 holds, where p(N) is the probability of obtaining e≤τ. In order to achieve the above condition, the threshold number $N^{*}$ of grid nodes is required, where $p(N^{*})=1$. Determining the threshold number of grid nodes $N^{*}$ is an important process in order to utilise the probabilistic approach, which depends on computations of random variables with the deterministic approach based on the results obtained from the sampling procedure.

As a result of the above conclusions, the classification of computational grid nodes can be redefined according to the threshold number of grid point $N^{*}$. For a spatial pattern of population density distributions, a grid is considered ultra-coarse if the number of grid nodes $N<N^{*}$, the transmission from ultra-coarse to coarse grids will be at $N=N^{*}$ and the error will be deterministic. Finally, a fine grid is defined as a grid in which $N>N^{*}$ and the asymptotic error (Equation 5) always holds. Thus, reliable approaches to evaluating the threshold number $N^{*}$ must be determined, especially as we are dealing with unknown features of spatial density distribution. In the numerical experiments, it has been proven that the estimation of the threshold number $N^{*}$ is a reliable means of determining the minimum number of grid nodes required to achieve sufficient accuracy.

On coarse grids, the locations of essential features of the spatial density distribution are not known in advance, which is an intrinsic property of the problem. Therefore, this paper has investigated the probability on a sample of coarse grid where a random location was provided for the first grid node, which reflects the lack of information about density patterns with several humps, patches or nonzero population densities. To deal with this uncertainty, the threshold number $N^{*}$ for reliable spatial distribution data, available in real-world settings, must be determined.

The analysis of the probabilistic approach was expanded to involve 2d problems. The 2d form of the normal distribution, with a varying number of patches and several values for the chosen tolerance, are considered with the purpose of comparing to the 1d problem. The critical number $N^{*}$ required to achieve an accurate estimate in 1d cases was considerably lower than the number required in 2d cases for the condition p(N)=1 to hold. Furthermore, it has been demonstrated that the value of the critical number $N^{*}$ depends on the number of patches, where for a small number of patches, $N^{*}$ increases, but it decreases gradually when the number of patches *P* increases from P=1 to P=8. The width of each patch and the values of the chosen tolerance τ obviously affect the required value of the critical number $N^{*}$. It has been explained in this paper that with a wide patch, $N^{*}$ becomes smaller than $N^{*}$ at narrow patch. The required number $N^{*}$ relies on the value of the chosen tolerance τ, where at a low level of accuracy (i.e., at a large value of τ), $N^{*}$ is quite small; this number increases when greater precision is required. The process of slightly altering the position of the bottom left corner grid node $\left(x_{1},y_{1}\right)$ within a limited domain $\left[0,h_{1}\right]\times \left[0,h_{2}\right]$ in the application of 2d ecological distributions is beneficial and highly recommended for complicated distribution pattern. In a 2d ecological distribution, we often do not have any prior information about the properties of the pattern and the location of the domain of non-zero density. The optimal positions of the grid nodes are difficult to discern, therefore the choice of point *A* will be random. In other words, at each random location of grid *A*, the evaluation of the total population size is considered as a single realisation of the random variable $I_{a}$, and different position of *A* will result in different values of $I_{a}$. Let us emphasis that, the location of *A* is not important on a sampling of very fine grid nodes, because uncertainty about the optimal location of grid nodes no longer applies.It is worth to mention in our paper that several studies applied the probabilistic approach to deal with sampling of coarse grid. It was suggested in [9] that the total population size on coarse sampling grids is a random variable and it has to be treated by applying probabilistic approaches. It was introduced in [9] the concept of ultra-coarse grids and the definition of an ultra-coarse grid implies that the probability *p* of achieving an integration error smaller than the given tolerance rather than to evaluate the error itself. The approach suggested in [9] the minimum number $N^{*}$ must be obtained to achieve desirable accuracy of integration which is guaranteed on a grid of $N^{*}$ nodes. In addition, it confirmed that the uniform refinement of an ultra-coarse grid will not reduce the value of integration error, on contrary the integration error can be even bigger than the error on the original grid unless the number $N^{*}$ of nodes is reached. Moreover, approximately the same result was obtained in [25], and this issue was discussed and introduced reliability criterion required to reconcile the probabilistic and deterministic approaches when the total population size is evaluated. These mentioned studies are supported to our results in this paper.

## Figures and Tables

**Figure 1 entropy-22-00658-f001:**
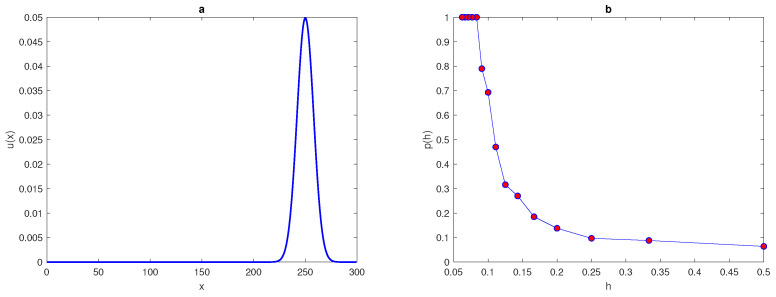
(**a**) A peak function (Equation 7). The peak width is δ=6σ where σ=8. (**b**) The probability p(h) obtained by direct computation for Function (Equation 7). δ is peak width, σ is standard deviation, *p* is the probability of achieving a sufficient accuracy, and *h* is the grid step size.

**Figure 2 entropy-22-00658-f002:**
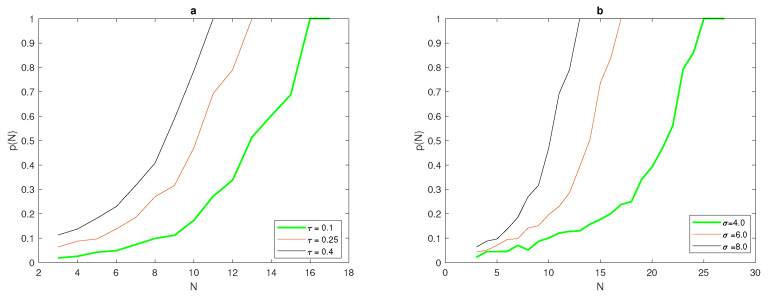
(**a**) Graphs of the probability function p(N) for fixed σ=8 in (Equation 7) and various values of tolerance τ. (**b**) Graphs of the probability function p(N) for fixed τ=0.25 in (Equation 7) and various values of parameter σ. σ is standard deviation, $N^{*}$ is the threshold number of grid nodes required to achieve probability condition, and τ is the accuracy tolerance.

**Figure 3 entropy-22-00658-f003:**
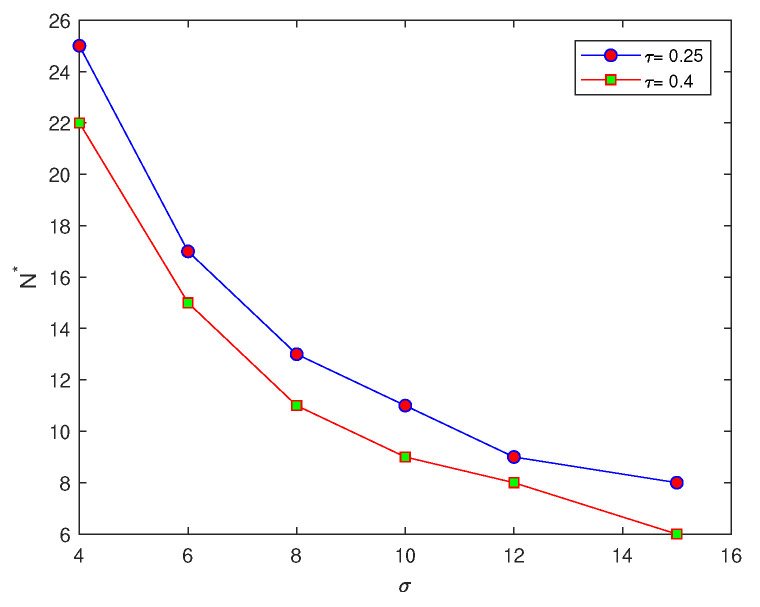
The critical number of grid nodes for the probabilistic approach at different values of σ and different values of τ. σ is standard deviation, *p* is the probability of achieving a sufficient accuracy, *N* is the number of grid nodes, and τ is accuracy tolerance.

**Figure 4 entropy-22-00658-f004:**
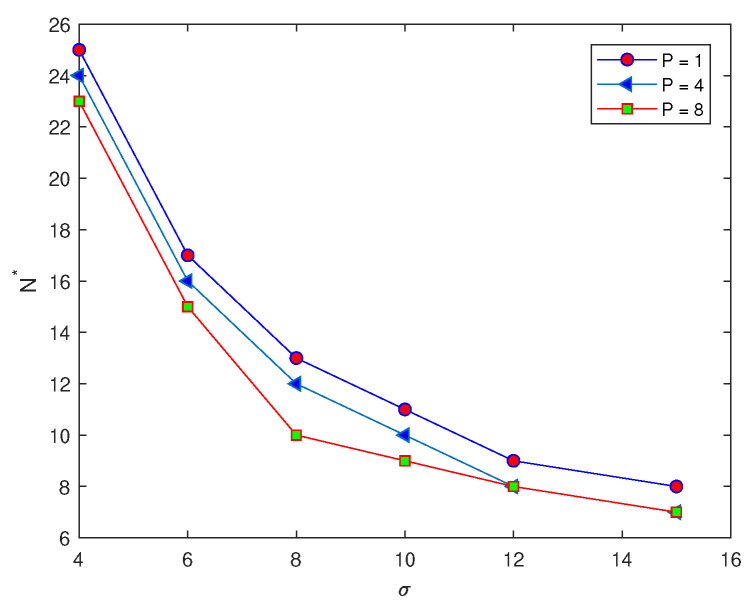
The number of grid nodes required to achieve an accurate estimate of the function with different peaks P=1,4,8 at a fixed value of tolerance τ=0.25. σ is standard deviation, $N^{*}$ is the threshold number of grid nodes required to achieve probability condition, and τ is the accuracy tolerance.

**Figure 5 entropy-22-00658-f005:**
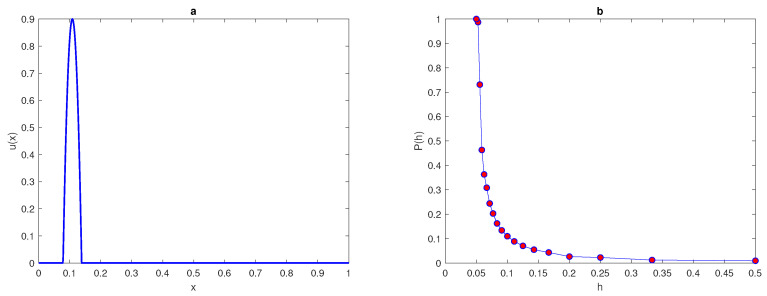
(**a**) A spatial population density given by quadratic function (Equation 14). The peak width is δ=0.06 and the parameters are A=1000 and B=0.9. (**b**) The probability p(h) was obtained by direct computation for the function (Equation 14) at δ=0.06. δ is the width of a peak, σ is standard deviation, *p* is the probability of achieving a sufficient accuracy, and *h* is the grid step size.

**Figure 6 entropy-22-00658-f006:**
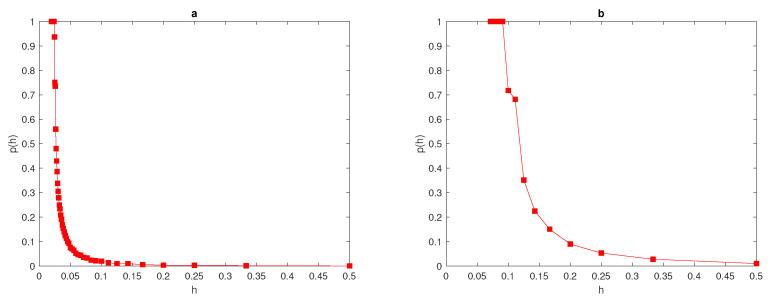
(**a**) The probability p(h) obtained by direct computation of the function (Equation 14) at δ=0.03. (**b**) The probability p(h) obtained by direct computation of the function (Equation 14) at δ=0.12. δ is the width of a peak, *p* is the probability of achieving a sufficient accuracy, and *h* is the grid step size.

**Figure 7 entropy-22-00658-f007:**
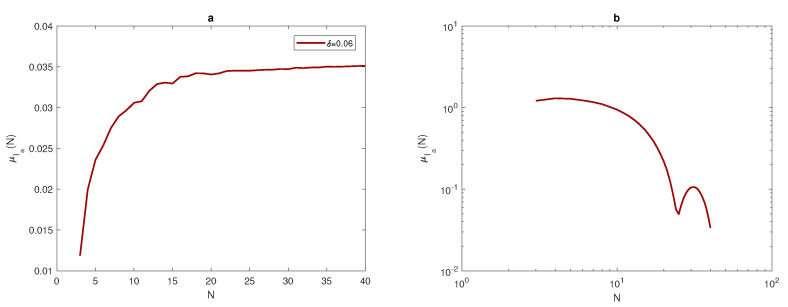
(**a**) The arithmetic mean for $I_{a}$ at δ=0.06. (**b**) The arithmetic mean for *e* at δ=0.06. δ is the width of a peak, μ is the arithmetic mean, $I_{a}$ is the approximate value of integral, *e* is the relative error, and *N* is the number of grid nodes.

**Figure 8 entropy-22-00658-f008:**
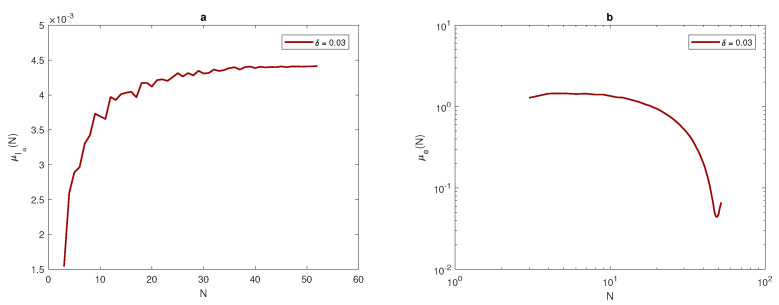
(**a**) The arithmetic mean for $I_{a}$ at δ=0.03. (**b**) The arithmetic mean for *e* at δ=0.03. δ is the width of a peak, μ is the arithmetic mean, $I_{a}$ is the estimate of pest abundance, *e* is the relative error, and *N* is the number of grid nodes.

**Figure 9 entropy-22-00658-f009:**
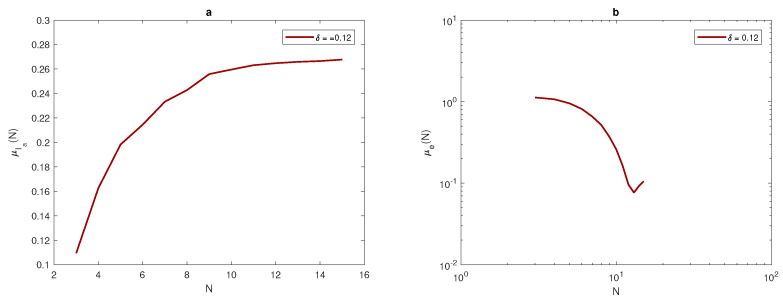
(**a**) The arithmetic mean for $I_{a}$ at δ=0.12. (**b**) The arithmetic mean for *e* at δ=0.12. δ is the width of a peak, μ is the arithmetic mean, $I_{a}$ is the approximate value of integral, *e* is the relative error, *N* is the number of grid nodes.

**Figure 10 entropy-22-00658-f010:**
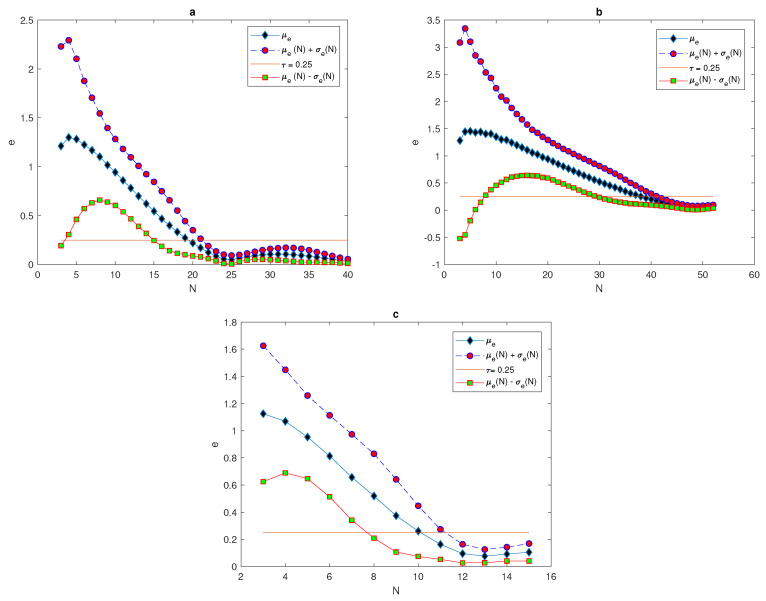
Analysis of the evaluation error for the quadratic function (Equation 14): (**a**) The function $\mu_{e}\left(N\right)$ and $\mu_{e}\left(N\right)\pm \sigma_{e}\left(N\right)$ at δ=0.06. (**b**) The function $\mu_{e}\left(N\right)$ and $\mu_{e}\left(N\right)\pm \sigma_{e}\left(N\right)$ at δ=0.03. (**c**) The function $\mu_{e}\left(N\right)$ and $\mu_{e}\left(N\right)\pm \sigma_{e}\left(N\right)$ at δ=0.12. δ is the width of a peak, μ is the arithmetic mean, *e* is the relative error, *N* is the number of grid nodes, σ is the standard deviation, and τ is the accuracy tolerance.

**Figure 11 entropy-22-00658-f011:**
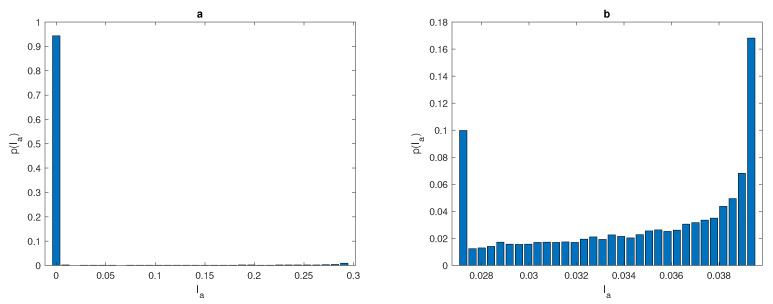
The probability of having the value $I_{a}$ of sub-intervals $\left[I_{a}\left(i\right),I_{a}\left(i+1\right)\right]$ for the spatial population density distribution (Equation 14) at δ=0.06 when (**a**) a coarse grid of N=3 points and (**b**) a fine grid of N=23 points. *p* is the probability of achieve a sufficient accuracy, δ is the width of a peak, *N* is the number of grid nodes, and $I_{a}$ is the estimate of pest abundance.

**Figure 12 entropy-22-00658-f012:**
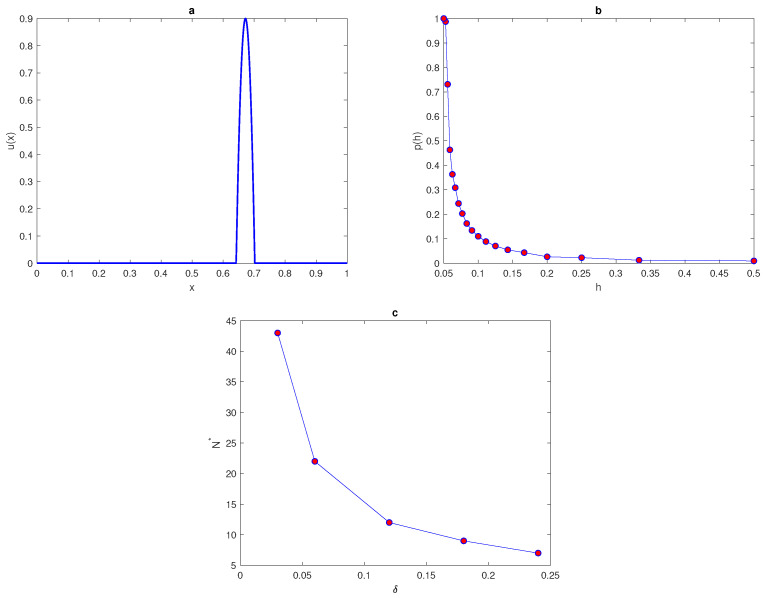
(**a**) The quadratic peak function (Equation 14) at δ=0.06 and $x^{*}=0.672$. (**b**) The probability p(h) obtained by direct computation for the function (Equation 14) at δ=0.06 and $x^{*}=0.672$. (**c**) The number of grid nodes required for the condition p(N)=1 to hold for the function (Equation 14) at a fixed location of $x^{*}=0.672$ and different values of peak widths δ. *p* is the probability of achieve a sufficient accuracy, δ is the width of a peak, *N* is the number of grid nodes, $x^{*}$ is the fixed location of the maximum of peak, and $N^{*}$ is the threshold number of grid nodes to hold probability condition.

**Figure 13 entropy-22-00658-f013:**
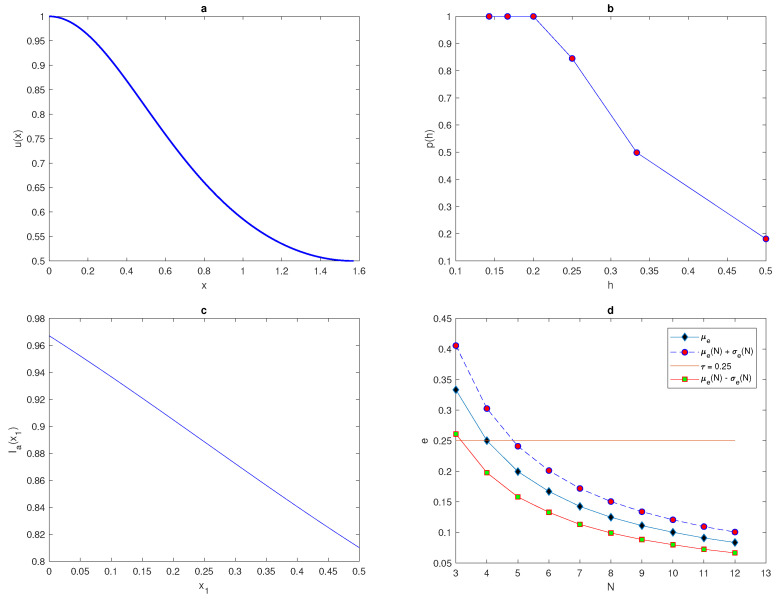
(**a**) The spatial population density distribution (Equation 25). (**b**) The probability p(h) obtained by direct computation for the function (Equation 25). (**c**) The function $I_{a}\left(x_{1}\right)$ generated on a uniform grid of N=3 nodes. (**d**) Analysis of the evaluation error for the spatial density distribution (Equation 25), where the mean of errors $\mu_{e}$, $\mu_{e}\left(N\right)+\sigma_{e}\left(N\right)$ and $\mu_{e}\left(N\right)-\sigma_{e}\left(N\right)$ are computed on a sequence of regular grids. μ is the arithmetic mean, *e* is the relative error, *N* is the number of grid nodes, σ is the standard deviation, τ is the accuracy tolerance, $x_{1}$ is the random location of the first node in the grid, and $I_{a}$ is the estimated pest abundance.

**Figure 14 entropy-22-00658-f014:**
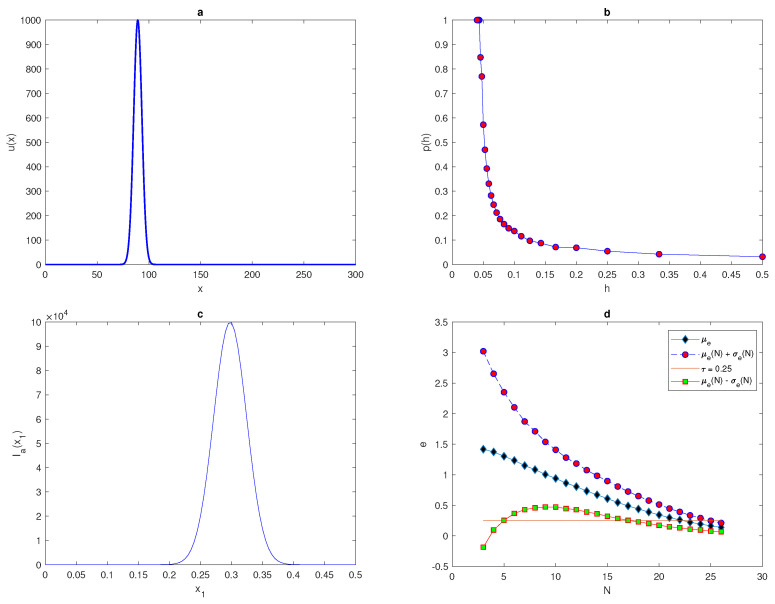
(**a**) The spatial population density distribution (Equation 27) at $x^{*}=89.3$. (**b**) The probability p(h) obtained by direct computation for the function (Equation 27). (**c**) The function $I_{a}\left(x_{1}\right)$ generated on a uniform grid of N=3 nodes. (**d**) Analysis of the evaluation error for the spatial density distribution (Equation 27), where the mean of errors $\mu_{e}$, $\mu_{e}\left(N\right)+\sigma_{e}\left(N\right)$ and $\mu_{e}\left(N\right)-\sigma_{e}\left(N\right)$ are computed on a sequence of regular grids. μ is the arithmetic mean, *e* is the relative error, *N* is the number of grid nodes, σ is the standard deviation, τ is the accuracy tolerance, $x_{1}$ is the random location of the first node in the grid, and $I_{a}$ is the estimate of pest abundance.

**Figure 15 entropy-22-00658-f015:**
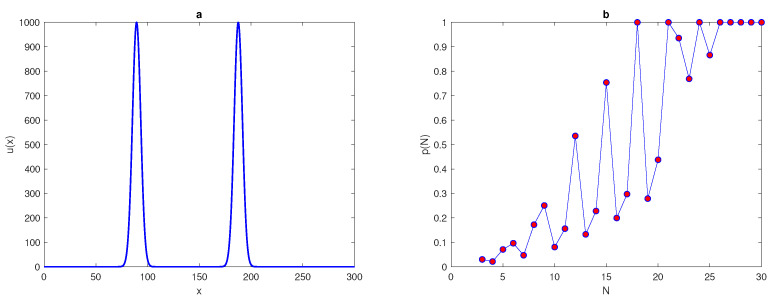
(**a**) The spatial population density distribution (Equation 28) at P=2, where A=1000, δ=0.06. (**b**) The probability p(h) obtained by direct computation for the function (Equation 28) at P=2. δ is the width of a peak, τ is the accuracy tolerance, *p* is the probability of achieve a sufficient accuracy, *N* is the number of grid nodes, and *P* is the number of peaks.

**Figure 16 entropy-22-00658-f016:**
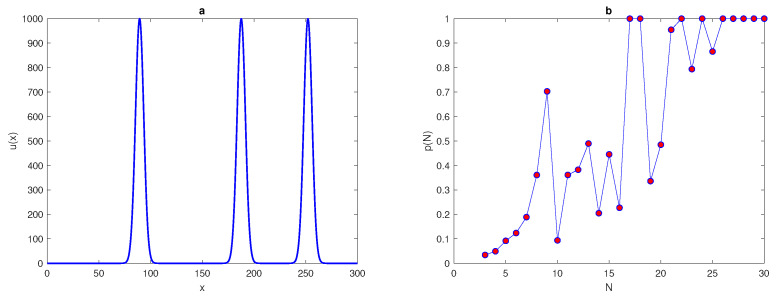
(**a**) The spatial population density distribution (Equation 28) at P=3, where A=1000, δ=0.06. (**b**) The probability p(h) obtained by direct computation for the function (Equation 28) at P=3. δ is the peak width, τ is the accuracy tolerance, *p* is the probability of achieve a sufficient accuracy, *N* is the number of grid nodes, and *P* is the number of peaks.

**Figure 17 entropy-22-00658-f017:**
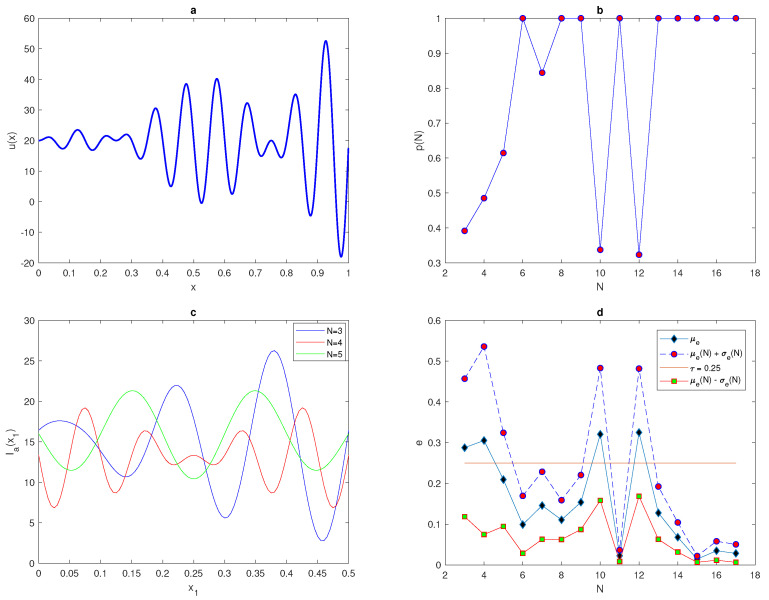
(**a**) The spatial population density distribution (Equation 29). (**b**) The probability p(h) obtained by direct computation for the function (Equation 29). (**c**) The function $I_{a}\left(x_{1}\right)$ generated on a uniform grid of N=3,4,5 nodes. (**d**) Analysis of the evaluation error for the spatial density distribution (Equation 29) where the mean of errors $\mu_{e}$, $\mu_{e}\left(N\right)+\sigma_{e}\left(N\right)$ and $\mu_{e}\left(N\right)-\sigma_{e}\left(N\right)$ are computed on a sequence of regular grids. μ is the arithmetic mean, *e* relative error, *N* is the number of grid nodes, σ is the standard deviation, τ is the accuracy tolerance, $x_{1}$ is the random location of the first node in the grid, and $I_{a}$ is the estimate of pest abundance.

**Figure 18 entropy-22-00658-f018:**
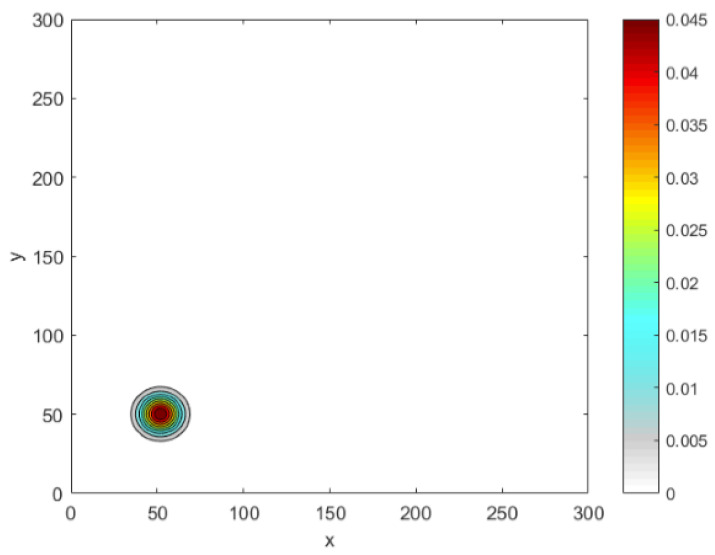
The spatial population distribution given by the 2d normal distribution (Equation 30) at fixed δ=0.06. δ peak width.

**Figure 19 entropy-22-00658-f019:**
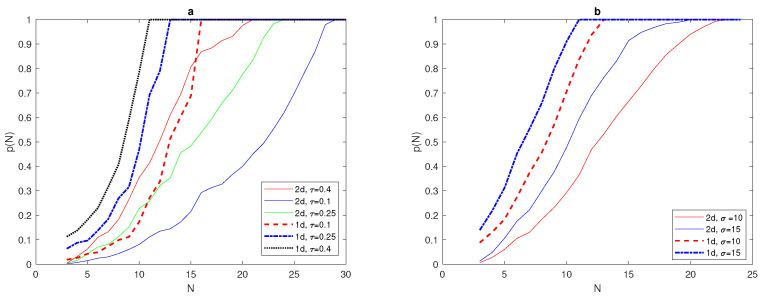
(a) The probability p(N) obtained by direct computation for functions (Equation 7) and (Equation 30) at fixed δ=0.06 and different values of tolerance τ = 0.1, 0.25, 0.4. (b)The probability p(N) obtained by direct computation for functions (Equation 7) and (Equation 30) at fixed τ=0.25 and different values of σ = 10, 15. *N* is the number of grid nodes, σ is the standard deviation, τ is the accuracy tolerance, and *p* is the probability of achieving sufficient accuracy.

**Figure 20 entropy-22-00658-f020:**
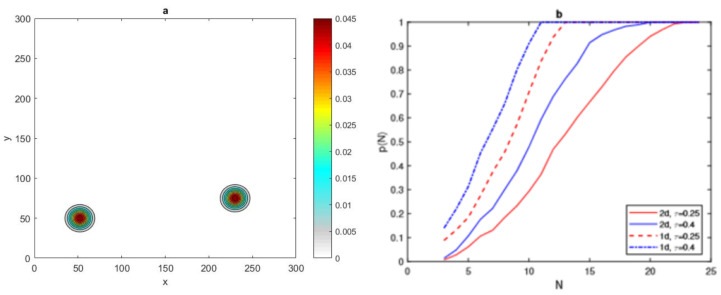
(**a**) The spatial population distribution given by a 2d normal distribution (Equation 31) at fixed δ=0.06 and P=2. (**b**) The probability p(N) obtained by direct computation for functions (Equation 31) and (Equation 11) at fixed δ=0.06 and different values of τ = 0.25, 0.4. *N* is the number of grid nodes, σ is the standard deviation, τ is the accuracy tolerance, *p* is the probability of achieving sufficient accuracy, and *P* is the number of peaks.

**Figure 21 entropy-22-00658-f021:**
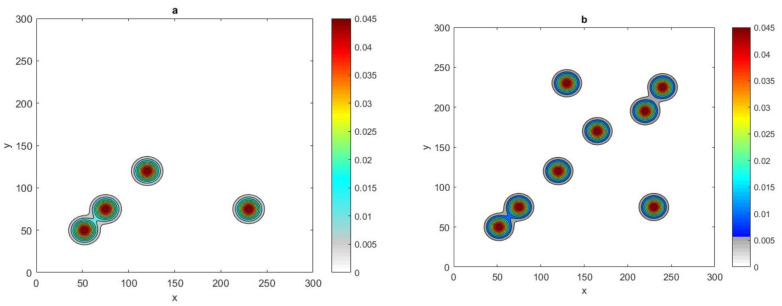
(**a**) The spatial population distribution given by a 2d normal distribution (Equation 31) at fixed δ=0.06 and P=4. (**b**) The spatial population distribution given by a 2d normal distribution (Equation 31) at fixed δ=0.06 and P=8. δ is the peak width and *P* is the number of peaks.

**Figure 22 entropy-22-00658-f022:**
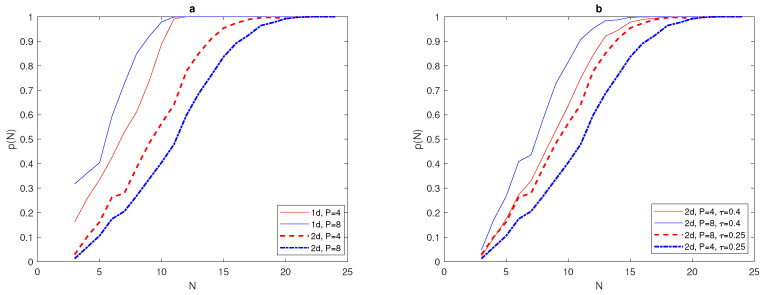
(**a**) The probability p(N) obtained by direct computation for functions (Equation 31) and (Equation 11) at fixed δ=0.06 and fixed tolerance τ=0.25, with different numbers of patches *P* = 4, 8. (**b**) The probability p(N) obtained by direct computation for function (Equation 31) at fixed δ=0.06, with different values of τ = 0.25, 0.4 and different numbers of patches P=4, 8. *N* is the number of grid nodes, σ is the standard deviation, τ is the accuracy tolerance, *p* is the probability of achieving sufficient accuracy, and *P* is the number of peaks.

**Figure 23 entropy-22-00658-f023:**
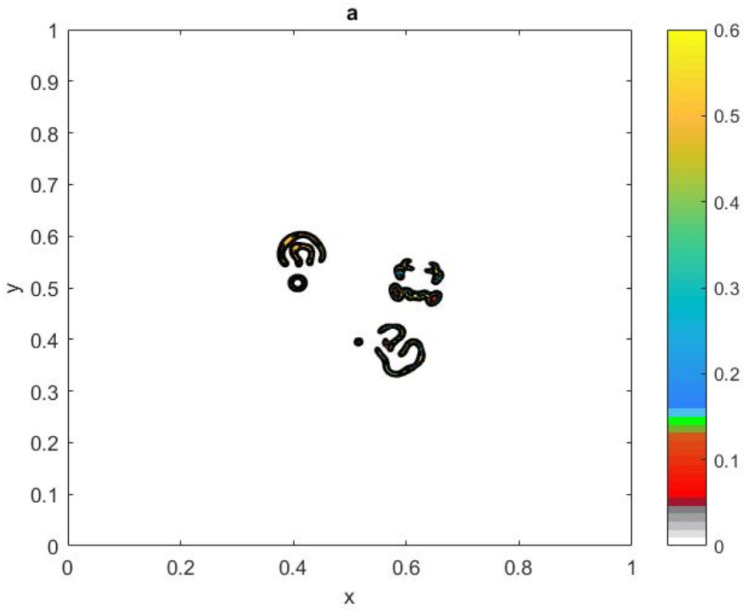
A highly aggregated spatial density distribution.

**Figure 24 entropy-22-00658-f024:**
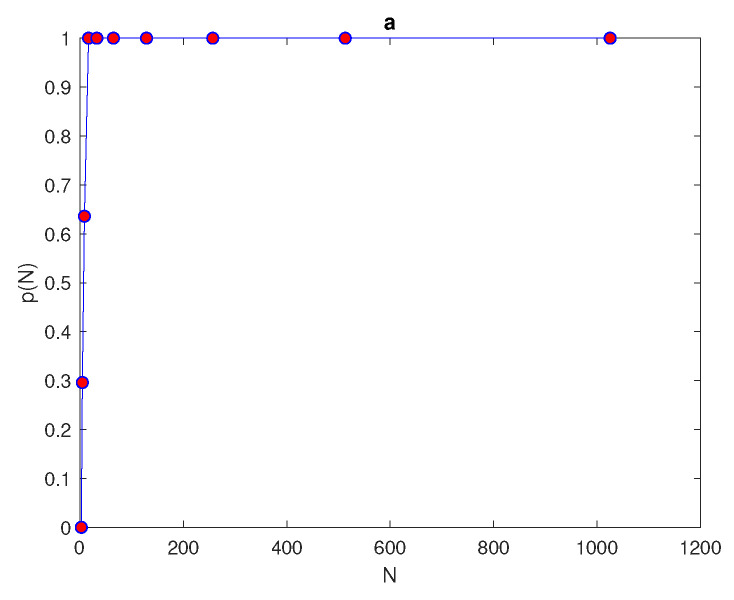
The probability p(N) obtained by direct computation for the patchy invasion spatial density distribution, with random locations of $\left(x_{1},y_{1}\right)$ at fixed tolerance τ=0.25. *N* is the number of grid nodes, τ is the accuracy tolerance, and *p* is the probability of achieving sufficient accuracy.

**Figure 25 entropy-22-00658-f025:**
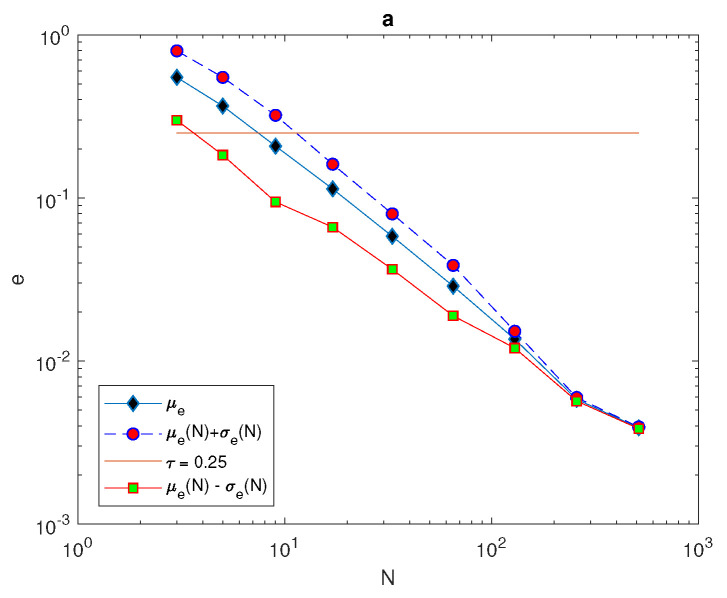
Analysis of the evaluation error for 2d spatial density distributions where the mean of errors $\mu_{e}$, $\mu_{e}\left(N\right)+\sigma_{e}\left(N\right)$ and $\mu_{e}\left(N\right)-\sigma_{e}\left(N\right)$ are computed on a sequence of regular grids for a patchy invasion spatial density distribution. μ is the arithmetic mean, *e* is the relative error, *N* is the number of grid nodes, σ is the standard deviation, τ is the accuracy tolerance, $x_{1}$ is the random location of the first node in the grid, and $I_{a}$ is the estimated of pest abundance.

**Table 1 entropy-22-00658-t001:** The number of grid nodes required to achieve an accurate estimate of the peak function (Equation 7) at fixed τ=0.25,0.4 and several values of σ.

σ	4	6	8	10	12	15
$N^{*}$ at τ=0.25	25	17	13	11	9	8
$N^{*}$ at τ=0.4	22	15	11	9	8	6

**Table 2 entropy-22-00658-t002:** The number of grid nodes required to achieve an accurate estimate of the peak function P=1,4,8 at fixed τ=0.25 and several values of σ.

σ	$N^{*}$ at P=1	$N^{*}$ at P=4	$N^{*}$ at P=8
4	25	24	23
6	17	16	15
8	13	12	10
10	11	10	9
12	9	8	8
15	8	7	7

**Table 3 entropy-22-00658-t003:** The number of grid nodes required to achieve an accurate estimate of the population density given by peak function (Equation 14), at fixed τ=0.25 and several values of δ alongside values of grid step size *h* where p(N)=1.

δ	$N^{*}$	*h*
0.03	43	$h=0.0238>\frac{\delta }{2}$
0.06	23	$h=0.0455>\frac{\delta }{2}$
0.12	12	$h=0.0909>\frac{\delta }{2}$

**Table 4 entropy-22-00658-t004:** The number of grid nodes required to achieve an accurate estimate of the peak function (Equation 14) at fixed τ=0.25 and several values of δ alongside the number of grid nodes required for the condition (Equation 23) to hold.

δ	p=1	$\mu_{e}\leq 0.25$
0.03	$N^{*}$ = 43	$N^{*}$ = 39
0.06	$N^{*}$ = 23	$N^{*}$ = 20
0.12	$N^{*}$ = 12	$N^{*}$ = 11

**Table 5 entropy-22-00658-t005:** The probability p(N), the arithmetic mean ${\hat{\mu }}_{I_{a}}$, the probabilistic mean $\mu_{I_{a}}$, the difference $|I-\mu_{I_{a}}|$, and the probabilistic standard deviation $\sigma_{I_{a}}$ computed on a sequence of regular computational grid nodes for quadratic function (Equation 14) at δ=0.06,A=1000, and I=0.0360.

*N*	3	4	5	…	20	21	22	23	24
p(N)	0.0019	0.0067	0.0129	...	0.7151	0.7292	0.9877	1	1
${\hat{\mu }}_{I_{a}}$	0.0133	0.0196	0.0239	...	0.0341	0.0342	0.0343	0.0344	0.0345
$\mu_{I_{a}}$	0.0176	0.0225	0.0261	...	0.0342	0.0343	0.0344	0.0359	0.0360
$\left|I-\mu_{I_{a}}\right|$	0.0184	0.0135	0.0099	...	0.0018	0.0017	0.0016	0.0001	0
$\sigma_{I_{a}}$	0.2257	0.2249	0.2104	...	0.0376	0.0286	0.0211	0.0167	0.0102

**Table 6 entropy-22-00658-t006:** The number of grid nodes required to achieve an accurate estimate of the peak function (Equation 14) at fixed τ=0.25 and several values of δ alongside the number of grid nodes required to hold the condition p(N)=1.

δ	p=1 at Fixed $x^{*}$
0.03	$N^{*}$ = 43
0.06	$N^{*}$ = 22
0.12	$N^{*}$ = 12
0.18	$N^{*}$ = 9
0.24	$N^{*}$ = 7

**Table 7 entropy-22-00658-t007:** The number of grid nodes required to achieve an accurate estimate of the normal distribution functions (Equation 7) and (Equation 30) at several values of τ=0.25,0.4 and several values of σ, alongside the number of grid nodes required for the condition p(N)=1 to hold for both 1d and 2d problems.

σ	τ=0.25	τ=0.4
4	$N_{1d}^{*}$ = 25, $N_{2d}^{*}$ = 47	$N_{1d}^{*}$ = 22, $N_{2d}^{*}$ = 41
6	$N_{1d}^{*}$ = 17, $N_{2d}^{*}$ = 31	$N_{1d}^{*}$ = 15, $N_{2d}^{*}$ = 28
8	$N_{1d}^{*}$ = 13, $N_{2d}^{*}$ = 24	$N_{1d}^{*}$ = 11, $N_{2d}^{*}$ = 21
10	$N_{1d}^{*}$ = 11, $N_{2d}^{*}$ = 20	$N_{1d}^{*}$ = 9, $N_{2d}^{*}$ = 18
12	$N_{1d}^{*}$ = 9, $N_{2d}^{*}$ = 16	$N_{1d}^{*}$ = 8, $N_{2d}^{*}$ = 15
15	$N_{1d}^{*}$ = 8, $N_{2d}^{*}$ = 13	$N_{1d}^{*}$ = 6, $N_{2d}^{*}$ = 12

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
