# Peer review of "Using Probabilistic Approach to Evaluate the Total Population Density on Coarse Grids"

_entropy, 2020, doi:10.3390/e22060658_

Round 1

Reviewer 1 Report

The paper deals with the Using to evaluation of the total
population density on coarse grids with the use of probabilistic methods.

It is very interesting and the used methodology, results and conclusions are correct.

I have some minor comments:

  1. In each presented equations all used symbols must be described in the text (especially in Introduction).
  2. In each presented figure - the symbols on axes are only letters - they should be shown as full names of presented values - in present form the reader  has to find in the text what is presented in the figure.
  3. Also in the figures captions all used symbols should be presented with full names (not only symbols).
  4. The comment about the results correctness validation must be added - is there any data (literature, experimental etc.) that can proof the correctness of the presented study.

Author Response

Reviewer # 1

Thank you, Dear Reviewer, for your careful and thorough reading of our submission. Your help in this substantial and significant improvement of our paper is indeed greatly appreciate.

[1] In each presented equations all used symbols must be described in the text (especially in Introduction).

Reply: Thanks a lot for this suggestion. Done. Each equation has explained all symbols used and we add table of most repeated symbols align its description at the end of conclusion in section 8 please see page 38 in the revised version.

[2] In each presented figure - the symbols on axes are only letters - they should be shown as full names of presented values - in present form the reader has to find in the text what is presented in the figure.

Reply: Done. Please see the caption of each figure it is  rewritten in details as request in the revised version.

[3] The comment about the results correctness validation must be added - is there any data (literature, experimental etc.) that can proof the correctness of the presented study.

Reply: Done. Please see the highlighted text in the last paragraph in conclusion pages 36-37  in the revised version.

Reviewer 2 Report

The paper addresses the topic of evaluating the population density of ecological/biological systems with accuracy. The adopted techniques ar simple and well-known, but the paper has still the merit of the area of application. Therefore, the reviewer suggests extending slightly the introduction to the topic so that general readers can follow to application more easily. In general the writing style is adequate and the review of the literature covers relevant works. The reviewer recommends mainly the improvement of 2-dim charts with experimental/numerical data with a limited number of points. It would improve the paper to have a larger number of data points. In the same line of thought, the contour charts in several figures (e.g., 18, 20,21, 23) some change win the colour scale (e.g., minimum different from zero) would improve  the readability of the plots.

The impulses for small values of I_a in the plots of fig 11 deserve a better treatment (discussion, eventually a different scale, ...)

Some small details:

  • use \exp instead of exp
  • - a small lost dot "." in the caption of fig 8
  •  Titles of sections 5, 6, use "The 2d..." instead of "2d..."

Author Response

Thank you, Dear Reviewer, for your careful and thorough reading of our submission. Your help in this substantial and significant improvement of our paper is indeed greatly appreciate.

Reviewer # 2

 [1] The reviewer suggests extending slightly the introduction to the topic so that general readers can follow to application more easily.

Reply: Thanks a lot for this observation. It has been done please see highlighted paragraph in the introduction page 4 in revised version.

 [2] The reviewer recommends mainly the improvement of 2-dim charts with experimental/numerical data with a limited number of points. It would improve the paper to have a larger number of data points.

Reply: Thanks a lot for your recommendation, however we used a vast array of 2d data and we achieved desired accuracy and the probability condition is acquired, therefore increasing the number of grid nodes will not provide any new results.  

[3] In the same line of thought, the contour charts in several figures (e.g., 18, 20,21, 23) some change win the color scale (e.g., minimum different from zero) would improve the readability of the plots.
Reply: Thanks a lot for this observation. Done please see mentioned figured after moderating pages 28-31-33-32 in revised version.

[4] The impulses for small values of I_a in the plots of fig 11 deserve a better treatment (discussion, eventually a different scale, some small details.
Reply: Thanks a lot for this observation. Done please see highlighted paragraph in section 4 page 16 in revised version.

[5] use \exp instead of exp- a small lost dot "." in the caption of fig 8, Titles of sections 5, 6, use "The 2d..." instead of "2d..."

Reply: Thanks a lot for this observation. Done please see figure 8, and see pages 26-29 in revised version.